# Accounting for Precipitation Asymmetry in a Multiplicative Random Cascades Disaggregation Model

Kaltrina Maloku[1], Benoit Hingray[1], and Guillaume Evin[1]

[1]Univ. Grenoble Alpes, CNRS, INRAE, IRD, Grenoble INP, IGE, 38000 Grenoble, France)

**Correspondence:** Kaltrina Maloku (kaltrina.maloku@univ-grenoble-alpes.fr)

**Abstract.**

Analytical Multiplicative Random Cascades (MRCs) are widely used for the temporal disaggregation of coarse-resolution precipitation time series. This class of models applies scaling models to represent the dependence of the cascade generator on the temporal scale and the precipitation intensity. Although determinant, the dependence on the external precipitation pattern is usually disregarded in the analytical scaling models. Our work presents a unified MRC modelling framework that allows the cascade generator to depend in a continuous way on temporal scale, precipitation intensity and a so-called precipitation asymmetry index.

Different MRC configurations are compared for 81 locations in Switzerland with contrasted climates. The added value of the dependence of the MRC on the temporal scale appears to be unclear, unlike what was suggested in previous works. Introducing the precipitation asymmetry dependence in the model leads to a drastic improvement of model performance for all statistics related to precipitation temporal persistence (wet/dry transition probabilities, lag-n autocorrelation coefficients, lengths of dry/wet spells). Accounting for precipitation asymmetry seems to solve this important limitation of previous MRCs.

The model configuration that only accounts for the dependence on precipitation intensity and asymmetry is highly parsimonious, with only five parameters, and provides adequate performances for all locations, seasons and temporal resolutions. The spatial coherency of the parameter estimates indicates a real potential for regionalisation and for further application to any location in Switzerland.

# 1 Introduction

Multi-decadal time series of sub-daily precipitation, hourly or even higher temporal resolution, are necessary for many applications, e.g. assessment of soil erosion (Römkens et al., 2002; Jebari et al., 2012) or of flash floods due to intense precipitation (Rafieeinasab et al., 2015; Liang et al., 2016), hydrological simulation of small catchments, such as those in mountainous regions (Sikorska and Seibert, 2018) or urban areas (Ochoa-Rodriguez et al., 2015; Cristiano et al., 2017). However, fine-scale precipitation data are scarce and usually cover limited periods of time, rarely predating the 80's (Segond et al., 2006; Jennings et al., 2010). In contrast, daily precipitation observations are widely spread worldwide and cover much longer periods, even going back to the middle of the 19th century in some cases. To take benefit from their much richer information, high-resolution precipitation data are often derived from coarse-resolution observations (especially daily) via some appropriate disaggregation process. For a given sequence of daily precipitation amounts, for instance, some disaggregation can be used to generate hourly precipitation scenarios, distributing the precipitation amount observed each day to its different hourly subdivisions. Disaggregation has been widely used to produce high-resolution time series scenarios from observed time series of daily precipitation, e.g. Molnar and Burlando (2005), or from low-resolution synthetic ones obtained in a first step with a so-called stochastic weather generator (e.g. Koutsoyiannis and Onof, 2001; Kang and Ramírez, 2010; Paschalis et al., 2014). Disaggregation models have also been used to generate high-resolution spatial precipitation (multi-site or spatial fields, see e.g. Mezghani and Hingray, 2009; Evin et al., 2018; Viviroli et al., 2022).

Many disaggregation models have been presented in the past (see for example Srikanthan and McMahon (2001) and Koutsoyiannis (2003) and the references within). A widely used method is the non-parametric method of fragments (MOF), where the disaggregation scheme for any target day is obtained from the high-resolution temporal (and spatial, if relevant) structure of a given analog day selected in the archive of observations (e.g. Mezghani and Hingray, 2009; Breinl and Di Baldassarre, 2019; Park and Chung, 2020; Acharya et al., 2022). By construction, MOF methods preserve the sub-daily patterns of precipitation and the intermittency properties within each day. An alternative disaggregation method is provided by multiplicative random cascades (MRCs), inspired by the statistical theory of turbulent fields (Schertzer and Lovejoy, 1987; Tessier et al., 1993). Because of the simplicity of both parameter estimation and simulation processes, MRCs have been widely used in the past for many applications in hydrology, for the point temporal disaggregation of precipitation time series (e.g. Menabde and Sivapalan, 2000; Pui et al., 2012; Pohle et al., 2018), or for the spatial-temporal disaggregation of precipitation fields (e.g. Seed et al., 1999; Rupp et al., 2012; Schleiss, 2020).

MRC models are usually implemented with a so-called branching number equal to 2. In such a case, the amount of precipitation at any time step is partitioned into two parts, attributed respectively to the first and second subdivision of this time step. The partition is repeated throughout the cascade levels until the final temporal resolution is achieved (Olsson, 1998; Molnar and Burlando, 2005; Rupp et al., 2009). In the so-called micro-canonical MRC models, the partition of precipitation is conservative. The precipitation amounts $R_1$ and $R_2$ attributed respectively to the first and second subdivisions of the considered time step (with precipitation amount $R_0$), are expressed as $R_1 = W_1 \cdot R_0$ and $R_2 = W_2 \cdot R_0$, where $W_1 + W_2 = 1$. For a given time step, the disaggregation can be determined by the so-called breakdown coefficient (BDC) assigned to the first subdivision: $W := W_1$

which can take the values 0 with probability $p_{01}$ (no precipitation is attributed to the first subdivision), 1 with probability $p_{10}$ ($R_0$ is fully attributed to the first subdivision) or any value between 0 and 1. When $W \in (0,1)$, $W^+ = W$ follows a statistical distribution with probability density function $f_{W^+}$.

The probabilities $p_{01}$, $p_{10}$ and the distribution $f_{W^+}$ define $\Gamma$, the statistical distribution of $W$, called the cascade generator. For a given location, they have been found to depend on different factors. They first depend on the temporal scale and on the precipitation intensity of the precipitation amount $R_0$ to disaggregate (e.g. Molnar and Burlando, 2005; Rupp et al., 2009). For instance, $p_{01}$ and $p_{10}$ tend to significantly decrease with precipitation intensity and to be larger for higher time resolution; they also often exhibit some seasonality (Molnar and Burlando, 2008). The cascade generator $\Gamma$ also depends on the so-called external pattern of precipitation, i.e. on the temporal sequence of precipitation amounts $R_{t-1}, R_t, R_{t+1}$ around the precipitation amount $R_t$ to disaggregate (Ormsbee, 1989; Olsson, 1998; Güntner et al., 2001). For instance, $p_{01}$ tends to be higher and $p_{10}$ smaller in the case of a so-called "ascending" precipitation pattern (when $R_{t-1} < R_t < R_{t+1}$) and, conversely, $p_{01}$ tends to be smaller and $p_{10}$ higher with "descending" patterns ($R_{t-1} > R_t > R_{t+1}$).

Different models have been proposed for the cascade generator $\Gamma$, i.e. for the estimation of $p_{01}$, $p_{10}$, and $f_{W^+}$ as a function of selected factors of variability. They are either empirical or analytical. Empirical models are usually obtained from the empirical cumulative distribution function (ECDF) of $W$ estimated from time series of high-resolution observed data for a discrete number of dependency configurations. Parameter estimation is usually performed on a seasonal basis by considering the dependency on the temporal scales of interests, and/or the dependency on the external pattern of precipitation (Olsson, 1998; Ormsbee, 1989; Güntner et al., 2001). The main drawback of empirical MRCs is the considerable number of parameters to be estimated, the large number of classes to be considered, and as a consequence, the lack of robustness of parameter estimates. In practice, some dependencies of the ECDFs are thus often ignored.

Besides empirical MRC models, analytical models aim to represent in a synthetic way the dependency of $p_{01}$, $p_{10}$ and of the distribution $f_{W^+}$ to important factors of variability. Most analytical models have focused on the dependency on the temporal scale and precipitation intensity. For instance, the symmetric Beta distribution has been extensively used to model the distribution $f_{W^+}$ (Menabde and Sivapalan, 2000). The single parameter $\alpha$ of the symmetric Beta distribution shows strong relationships with the temporal scale and the precipitation intensity, and can be modelled with simple scaling analytical laws (Molnar and Burlando, 2005; Paulson and Baxter, 2007; Rupp et al., 2009). Analytical models typically use symmetric distribution models. Nevertheless, a few exceptions are worth mentioning. McIntyre et al. (2016) use two parameters asymmetric Beta distribution, and consider four external pattern classes for the estimation of its parameters (the so-called ending, starting, enclosed, isolated classes). Hingray and Ben Haha (2005) use an asymmetric piecewise linear distribution function, where the asymmetry of the distribution is estimated from a local precipitation asymmetry index defined from the deterministic model of Ormsbee (1989). In both cases, it was found that accounting for the asymmetry in the BDC distribution improved the reproduction of statistics related to precipitation persistence and intermittency, which are known to be difficult to reproduce with MRC models (e.g. Rupp et al., 2009; Paschalis et al., 2012, 2014; Müller and Haberlandt, 2018; Pohle et al., 2018). However, these MRC approaches did not include scaling models to account for the dependency of the parameters to the temporal scale and precipitation intensity, resulting in a rather large number of parameters to be estimated.

In the present work, we present an analytical MRC modelling framework that allows the cascade generator $\Gamma$ to depend in a continuous way on temporal scales, intensity, and external pattern of precipitation. The possibility to merge in a single and unified analytical scaling framework all the principal $\Gamma$ dependencies allows a minimal number of parameters while combining 1) scaling relationships with temporal scales and intensity and 2) scaling dependency on the external pattern. This approach aims to improve the model's relevance and performance. The following questions are considered:

- To what extent a continuous index of local precipitation asymmetry can describe the way the cascade generator $\Gamma$ depends on the external pattern of precipitation?

- Is it possible to identify some scaling behaviour with respect to this asymmetry index and is it possible to propose an analytical relationship to model this scaling behaviour?

- What is the added value of including such an asymmetry dependency in the cascade generator $\Gamma$, especially with regard to statistics related to precipitation persistence and intermittency?

One important application of disaggregation models is for locations where only coarse-resolution data are available. In this case, the parameters of the model cannot be estimated from local data, as such data are missing, and are obtained with some regionalisation process based on data available from neighbouring and/or locations with similar precipitation regimes (e.g. Hingray et al., 2014). For such a work, analytical scaling MRCs are promising. They are very parsimonious which is expected to ease model regionalisation and increase model robustness.

However, introducing precipitation asymmetry dependence is at the expense of model parsimony. In this work, we thus consider different MRC models, of different complexity, to find, if relevant, a compromise between model performance and parsimony. If the cascade generator is known to depict different types of scaling dependencies, not all are necessarily required to achieve fair model performance. Accounting for temporal scale dependence is widely considered to be beneficial. However, to our knowledge, the corresponding gain in performance is questionable (no improvement in Rupp et al. (2009) and loss of performance in Molnar and Burlando (2005)). Introducing dependence to intensity was conversely often found to significantly improve it (e.g. Rupp et al., 2009; Paschalis et al., 2014).

An additional objective of the present work is to investigate the loss of performance obtained when the cascade generator $\Gamma$ disregards the dependency on temporal scales. In particular, a model configuration that only accounts for the dependence on precipitation intensity and asymmetry is assessed.

The paper is structured as follows. In Sect. 2, we introduce a precipitation asymmetry index used to model the dependency on asymmetry. Four different analytical MRC models are also presented in this section. They account for dependency on the temporal scale, precipitation intensity and/or precipitation asymmetry. The different models are used for the disaggregation of daily precipitation time series available for a large set of stations in Switzerland. Station locations and precipitation data are presented in Sect. 3. The models are evaluated on their ability to reproduce a number of characteristic statistics of precipitation at multiple temporal scales. Main performance evaluation results are presented in Sect. 4, while the interests of different model components are discussed in Sect. 5. Section 6 concludes.

## 2 Methods

As mentioned previously, the cascade generator $\Gamma$ used for the disaggregation of any precipitation amount $R_0$ is defined by the probabilities $p_{01}$, $p_{10}$ and the distribution of $W^+$ denoted by $f_{W^+}$. In the following, $f_{W^+}$ is modelled with a 2-parameter Beta distribution following e.g. McIntyre et al. (2016):

$$f_{W^+} = \frac{1}{B(\alpha_1, \alpha_2)} W^{+(\alpha_1 - 1)} (1 - W^+)^{(\alpha_2 - 1)}, \tag{1}$$

where the Beta function $B(\alpha_1, \alpha_2)$ is a normalisation constant. The two parameters $\alpha_1$ and $\alpha_2$ are related to the mean and the variance of the distribution as follows:

$$E[W^+] = \frac{\alpha_1}{\alpha_1 + \alpha_2}, \tag{2}$$

and

$$Var[W^+] = \frac{\alpha_1 \alpha_2}{(\alpha_1 + \alpha_2)^2 (\alpha_1 + \alpha_2 + 1)}. \tag{3}$$

Note that when $\alpha_1 = \alpha_2 = \alpha$, the distribution is symmetric,

$$E[W^+] = 0.5 \tag{4}$$

and

$$Var[W^+] = \frac{1}{4(2\alpha + 1)} \tag{5}$$

The MRCs compared in this study consider different ways to model their dependencies to the temporal scale, precipitation intensity, and precipitation asymmetry. Four models are compared:

- Model A accounts for the dependency on the temporal scale and precipitation intensity.

- Model B is a simplification of model A. It disregards the dependency on the temporal scale.

- Models $A^+$ and $B^+$ are refinements of models A and B, where the dependency to asymmetry is accounted for.

In the following, we describe the way the different models represent the scaling relationships for $p_{01}$, $p_{10}$, $\alpha_1$ and $\alpha_2$. We first present the MRC modelling framework of model A and the simplifications chosen for model B. Next, we introduce a precipitation asymmetry index further considered to account for the dependency of the cascade generator $\Gamma$ on asymmetry. Finally, we describe how models $A^+$ and $B^+$ introduce this dependency.

To account for the seasonality of precipitation characteristics in the region, models are estimated and evaluated on a seasonal basis. Seasons are defined as follows: winter (December, January, February (DJF)), spring (March, April, May (MAM)), summer (June, July, August (JJA)) and autumn (September, October, November (SON)).

## 2.1 MRC models without asymmetry: Models A and B

In models A and B, the dependency to precipitation asymmetry is disregarded and the distribution of the cascade generator is assumed to be symmetric. The probabilities $p_{01}$ and $p_{10}$ are thus assumed equal to each other and the distribution of the strictly positive BDCs, $f_{W^+}$, is modelled with a symmetric Beta distribution, i.e. $\alpha_1 = \alpha_2 = \alpha$.

The probabilities $p_{01}$ and $p_{10}$ are estimated from $p_x = 1 - p_{01} - p_{10}$, where $p_x$ is the non-zero subdivision probability, i.e. the probability that the precipitation amount is split into two non-zero amounts. As illustrated with data from the Zurich station in Figure 1a, $p_x$ generally increases with the intensity of the precipitation amount to disaggregate and is expected to increase for coarser temporal scales. In model A, following Rupp et al. (2009), the scaling dependency of $p_x$ on the temporal scale $\tau$ and precipitation intensity $I$ is modelled as follows:

$$p_x(I, \tau) = \frac{1}{2}\left(1 + \mathrm{erf}\left[\frac{\log(I) - \mu(\tau)}{\sqrt{2}\sigma(\tau)}\right]\right), \tag{6}$$

where erf is the error function defined as $\mathrm{erf}(x) = \frac{2}{\sqrt{\pi}}\int_0^x e^{-t^2}\,dt$ and where $\mu$ and $\sigma$ are assumed to be linearly dependent on the logarithm of the temporal scale:

$$\mu(\tau) = a_\mu \log(\tau) + b_\mu, \tag{7}$$

$$\sigma(\tau) = a_\sigma \log(\tau) + b_\sigma. \tag{8}$$

In Model B, the dependency on the temporal scale is disregarded. In this case $\mu$ and $\sigma$ are assumed constant and equation (6) simplifies to:

$$p_x(I) = \frac{1}{2}\left(1 + \mathrm{erf}\left[\frac{\log(I) - \mu}{\sqrt{2}\sigma}\right]\right). \tag{9}$$

The distribution $f_{W^+}$ in models A and B is a symmetric Beta distribution. Its shape is defined by the value of its unique parameter $\alpha$, which is related to the variance of $W^+$ via equation (5). For $\alpha = 1$, the probability density function (pdf) $f_{W^+}$ is constant between 0 and 1, i.e. it corresponds to a uniform distribution. For $\alpha > 1$, the larger the value of $\alpha$, the smaller the variance of $W^+$, the more the pdf is concentrated around its mean, 0.5. A roughly equal partition towards both sub-time steps becomes more likely in this case and leads to a stronger persistence of precipitation. For $\alpha < 1$, the smaller the value of $\alpha$, the larger the variance of $W^+$, the more the pdf is concentrated close to 0 and 1, which leads to more variable and/or intermittent precipitation. In autumn, for the Zurich station, Figure 1 shows that $\alpha$ increases ($Var[W^+]$ decreases) when precipitation intensity increases (Figure 1b) and $\alpha$ decreases ($Var[W^+]$ increases) when the temporal scale gets coarser (Figure 1b,c).

In model A, $\alpha$ is modelled as a function of precipitation intensity and temporal scale. Following Rupp et al. (2009), the model is the product of two functions of both variables, illustrated for Zurich data in Figure 1c and 1d respectively:

$$\alpha(\tau, I) = g(I) \cdot h(\tau), \tag{10}$$

where :

$$h(\tau) = \alpha_0 \cdot \tau^H, \tag{11}$$

and

$$\log [g(I)] = c_0 + c_1 \log(I) + c_2 [\log(I)]^2. \tag{12}$$

In model B, the dependency of $\alpha$ on the temporal scale is disregarded. The dependency on intensity is modelled with the following function:

$$\log(\alpha(I)) = \begin{cases} 0, & \text{if } I \leq I_0 \\ K [\log(I/I_0)]^2, & \text{if } I_0 < I \leq I_1 \\ K [\log(I_1/I_0)]^2, & \text{if } I > I_1 \end{cases} \tag{13}$$

where $I_0 = 0.1$ mm h$^{-1}$ and $I_1 = 10$ mm h$^{-1}$, illustrated for Zurich data in Figure 1b. The distribution $f_{W^+}$ is thus uniform for very low intensities. For large intensities, $\alpha$ is bounded to the value obtained for $I_1$. The higher the value for $K$, the higher the curvature of the scaling function $\log(\alpha(I))$. Positive $K$ values lead to higher $\alpha$ values for larger intensities (and vice versa). Models A and B have respectively nine ($a_\mu, b_\mu, a_\sigma, b_\sigma$ for $p_x$ and $\alpha_0, H, c_0, c_1, c_2$ for $f_{W^+}$) and three ($\mu, \sigma$ for $p_x$ and $K$ for $f_{W^+}$) invariant parameters to be estimated.

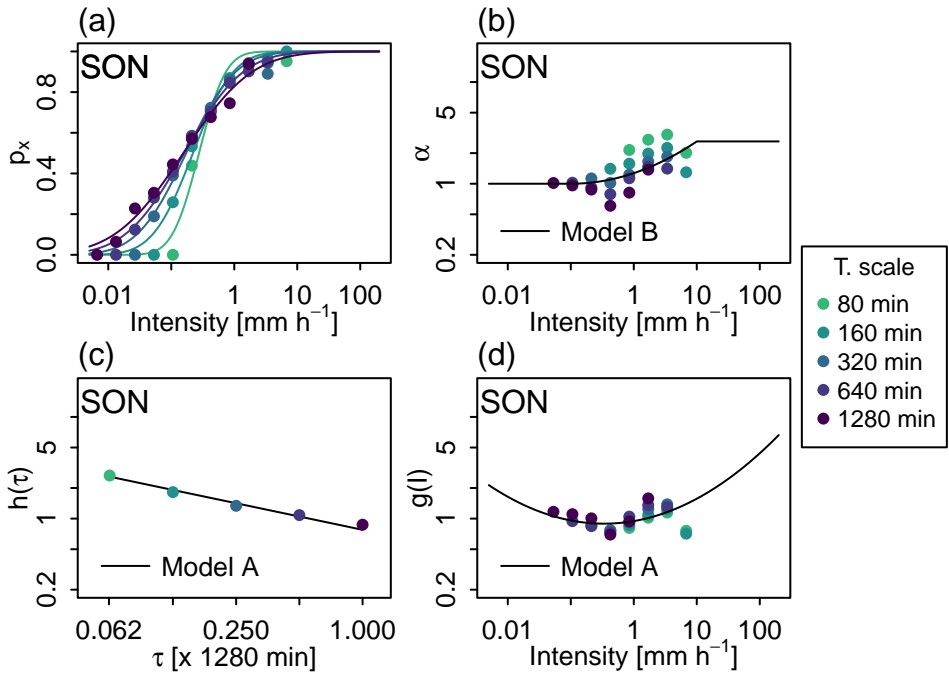

**Figure 1.** First row : two characteristics of the cascade generator $\Gamma$ as a function of precipitation intensity $I$ and temporal scale $\tau$. (a) non-zero subdivisions probability $p_x$ and (b) parameter $\alpha$ of the Beta distribution for $f_{W^+}$. In panel (a), the different lines correspond to the different $p_x(I)$ models obtained for different scales according to the scaling model of model A (equation 6). In panel (b), the line corresponds to the scaling model for $\alpha$ in model B (equation 13). In both graphs, dots correspond to empirical estimates used for model fitting. They are obtained for different classes of intensity and different temporal scales. The colour indicates the temporal scale. Second row : the two scaling sub-models for $\alpha$ in model A. (c) model $h(\tau)$ (equation 11) and (d) model $g(I)$ (equation 12). In panel (c), dots correspond to empirical estimates of $\alpha$ for the different temporal scales (all intensities included). In panel (d), dots correspond to empirical estimates of the ratio $\alpha(I,\tau)/h(\tau)$ for different classes of intensity and different temporal scales. Data from the Zurich station, autumn season.

## 2.2  An asymmetry index of precipitation sequences

In models $A^+$ and $B^+$, the cascade generator can be asymmetric: the probabilities $p_{01}$ and $p_{10}$ are not necessarily equal and the 2-parameter Beta distribution used to model $f_{W^+}$ can be asymmetric. As we will explain in the following, the asymmetry of the cascade generator is assumed to depend in a continuous way on the asymmetry of the precipitation sequence $\{R_{t-1}, R_t, R_{t+1}\}$. In other words, we assume that the larger the asymmetry of the sequence is, the larger the asymmetry of the cascade generator is expected to be.

To characterize the asymmetry of $\{R_{t-1}, R_t, R_{t+1}\}$, we introduce an asymmetry index $Z_t$ as in Hingray and Ben Haha (2005). $Z_t$ is defined as the hidden breakdown coefficient of the $\{R_{t-1}, R_t, R_{t+1}\}$ sequence. It is estimated from the two hidden precipitation amounts $R_1^* = R_{t-1} + 0.5R_t$ and $R_2^* = 0.5R_t + R_{t+1}$ that would be respectively obtained for the first and

second halves of the $\{R_{t-1}, R_t, R_{t+1}\}$ sequence if $R_t$ was split in half. $Z_t$ thus reads:

$$Z_t = \frac{R_{t-1} + 0.5R_t}{R_{t-1} + R_t + R_{t+1}}. \tag{14}$$

While $W$ is defined from the central precipitation and its subdivision amounts, $Z_t$ is fully determined from the central and its adjacent precipitation amounts. Details about the use of this index in the cascade generator are given in the following sections. $Z_t$ depends on the relative precipitation ratios $R_{t-1}/R_t$ and $R_{t+1}/R_t$ and varies from 0 to 1 as illustrated in Figure 2.

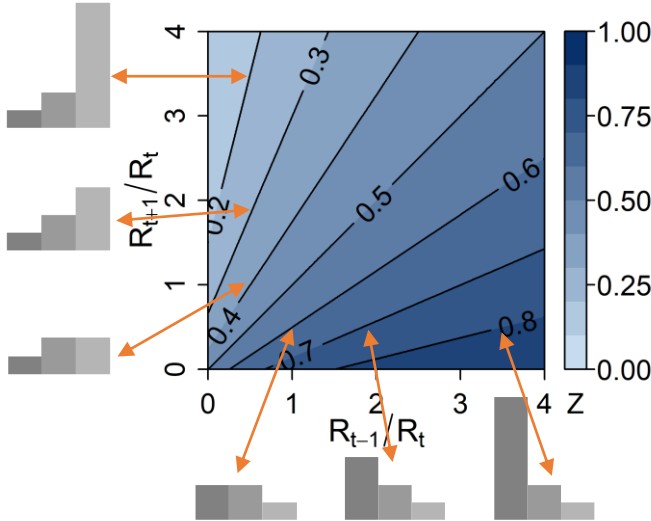

**Figure 2.** Asymmetry index $Z_t$ of the $\{R_{t-1}, R_t, R_{t+1}\}$ precipitation sequence as a function of the two precipitation ratios $R_{t-1}/R_t$ (x-axis) and $R_{t+1}/R_t$ (y-axis).

The following points can be noticed:

– A symmetric precipitation sequence, i.e. when $R_{t-1} = R_{t+1}$, leads to $Z_t = 0.5$ whatever the amount of precipitation on the central time step.

– For sequences with central precipitation amounts much larger than the adjacent ones, $Z_t$ is close to 0.5.

– $Z_t < 0.5$ indicates an "increasing" or "right valley" sequence, i.e. a sequence where $R_{t-1} < R_{t+1}$, while $Z_t > 0.5$ indicates a "decreasing" or "left valley" sequence ($R_{t-1} > R_{t+1}$).

– The $Z_t$ asymmetry index characterizes in a continuous way the "intensity of the asymmetry"; different "decreasing" precipitation sequences will have different $Z_t$ values depending on the "steepness" of the decrease. The larger the deviation from 0.5, the larger the asymmetry.

– $Z_t$ values close to 0 indicate sequences with very little rain on the first two time steps when compared to the last one (very steep "ascending" sequences), whereas $Z_t$ values close to 1 indicate sequences with very little rain on the two last time steps when compared to the first one (very steep "descending" sequences).

## 2.3 ECDF dependency on precipitation asymmetry and scaling models

Figure 3a shows the ECDFs of observed weights $W$ obtained for the Zurich station for 10 classes of precipitation asymmetry index, defined as $G_l^Z = \{W_t | Z_t \in (0.1(l-1), 0.1l]\}$, for $l = 1, \ldots, 10$. As expected, the ECDF depends a lot on the asymmetry class. The lower the value of $Z_t$ (i.e. corresponding to very steep ascending external patterns), the higher the value of $p_{01}$ and the lower the value of $p_{10}$. In other words, the higher the probability that all the $R_t$ precipitation amount is attributed to the second subdivision and the lower the probability that all $R_t$ is attributed to the first subdivision. The exact opposite happens for high values of $Z_t$ (i.e. values corresponding to very steep descending external patterns).

The asymmetry of the precipitation sequence directly translates to an asymmetry of the ECDF of $W$, namely to an asymmetry between $p_{01}$ and $p_{10}$ and to an asymmetry of the distribution $f_{W+}$. As shown in the following, the asymmetry of ECDF significantly depends on the value of the asymmetry index $Z$. For clarity, the temporal index $t$ is omitted hereafter.

To quantify the asymmetry between $p_{01}$ and $p_{10}$, we introduce the probability asymmetry ratio, $\varphi$, defined as:

$$\varphi = \frac{p_{01}}{p_{10} + p_{01}} = \frac{p_{01}}{1 - p_x}. \tag{15}$$

For $\varphi$ values close to 0.5, the asymmetry between $p_{10}$ and $p_{01}$ is small. The case $\varphi = 0.5$ corresponds to the symmetric case, i.e. $p_{10} = p_{01}$. The most asymmetrical configurations are obtained when $\varphi = 0$ or $\varphi = 1$. The case $\varphi = 0$ (resp. $\varphi = 1$) corresponds to $p_{01} = 0$ and $p_{10} = 1 - p_x$ (resp. $p_{01} = 1 - p_x$ and $p_{10} = 0$). The relationship between $\varphi$ and $Z$ is illustrated in Figure 3b. As expected, when $Z$ increases, $\varphi$ decreases. The $\varphi(Z)$ relationship is found to be roughly similar for all Swiss stations (results not shown) and can be modelled with an error function as follows:

$$\varphi(Z) = \frac{1}{2}\left(1 + \text{erf}\left(\frac{0.5 - Z}{\nu\sqrt{2}}\right)\right), \tag{16}$$

where the single parameter $\nu$ is related to the strength of the $\varphi(Z)$ relationship around the pivot point ($Z = 0.5$, $\varphi = 0.5$). The strength is found to depend on the station and on the season. It is for instance slightly higher in winter than in summer for Zurich station.

For asymmetric precipitation sequences, the distribution $f_{W+}$ is also expected to be asymmetric. In particular, the mean of the distribution is expected to differ from 0.5. This is illustrated in Figure 3c for Zurich observation data, where the mean of $W^+$, denoted by $m$, is estimated for different classes of $Z$. As expected, the lower the value of $Z$, the lower the value of the mean. In terms of precipitation dynamics, this reflects the fact that in an ascending precipitation sequence, the amount of precipitation tends to be smaller in the first time subdivision than in the second one. The reverse is true in the case of a descending sequence. The $m(Z)$ relationship highlighted in Figure 3c is found to be roughly linear for all Swiss stations (results not shown) and can be modelled as:

$$m(Z) = \lambda\left(Z - \frac{1}{2}\right) + \frac{1}{2}. \tag{17}$$

The model is centred on the pivot point ($Z = 0.5$, $m = 0.5$). Its unique parameter $\lambda$ corresponds to the slope of this linear function and indicates the strength of the dependency on precipitation asymmetry $Z$. The strength is also found to depend on the season and site.

In models $A^+$ and $B^+$ described in the following section, the dependency of the cascade generator $\Gamma$ on precipitation asymmetry is accounted for with the help of the scaling models $\varphi(Z)$ and $m(Z)$. Both scaling models have one single invariant parameter to estimate.

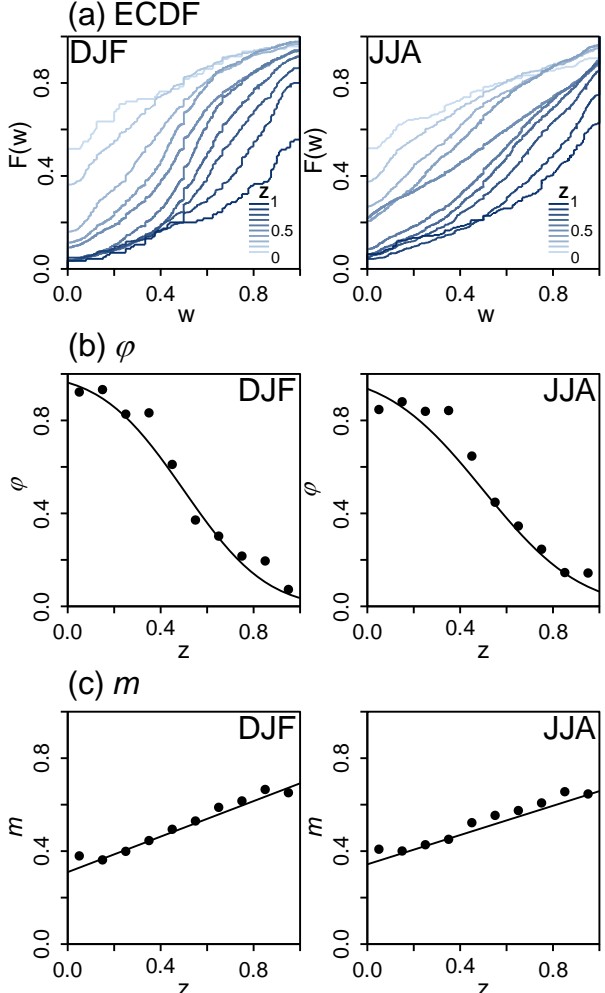

**Figure 3.** Statistical characteristics of the breakdown coefficients $W$ as a function of the asymmetry index $Z$ for Zurich station in winter (DJF, left column) and summer (JJA, right column). (a) ECDFs of $W$ for 10 classes of $Z$. (b) Dry probability asymmetry ratio $\varphi$ as a function of $Z$. The estimates $\varphi$ obtained for different classes of $Z$ are indicated with black dots and the fitted model $\varphi(Z)$ with a plain black line. (c) Estimated mean of $W^+$ as a function of $Z$. Empirical means are indicated with black dots and the fitted model $m(Z)$ with a plain black line.

## 2.4 MRC models with asymmetry : Models $A^+$ and $B^+$

The introduction of asymmetry in model A, leading to model $A^+$, is described hereafter. In model $A^+$, the non-zero subdivision probability $p_x$ is obtained for $I$ and $\tau$ as in model A with the scaling model $p_x(I, \tau)$ presented in Sect. 2.1 (see equation 6).

Contrary to model A, probabilities $p_{01}$ and $p_{10}$ can be different and are obtained from the definition (15) of the probability asymmetry ratio $\varphi(Z)$ as follows:

$$p_{01} = \varphi(Z)\,(1 - p_x)\,, \tag{18}$$

$$p_{10} = (1 - \varphi(Z))\,(1 - p_x)\,. \tag{19}$$

In model $A^+$, the Beta distribution $f_{W^+}$ can be asymmetric and its two parameters can be related to $E[W^+]$ and $Var[W^+]$ from equations (4) and (5) as follows:

$$\alpha_1 = \left( \frac{E[W^+]\,(1 - E[W^+])}{Var[W^+]} - 1 \right) E[W^+]\,, \tag{20}$$

$$\alpha_2 = \left( \frac{E[W^+]\,(1 - E[W^+])}{Var[W^+]} - 1 \right) \left(1 - E[W^+]\right). \tag{21}$$

In model $A^+$, we obtain $E[W^+]$ from the scaling model $m(Z)$ of equation (17) and $Var[W^+]$ is assumed to be the same as in model A, i.e. in a configuration where the distribution is assumed symmetric. In practice, $Var[W^+]$ is thus simply derived from the value of $\alpha$ obtained with the scaling model $\alpha(I,\tau)$ of model A as follows: $Var[W^+] = 1/(4\,(2\alpha(I,\tau)+1))$. The parameters $\alpha_1$ and $\alpha_2$ are then simply derived from these representations of $E[W^+]$ and $Var[W^+]$.

Model $B^+$ is derived from model B in the same way. The probabilities $p_{01}$ and $p_{10}$ are obtained from equations (18) and (19) with the scaling model $p_x(I)$ of equation (9). $\alpha_1$ and $\alpha_2$ are obtained from equations (20) and (21) where $E[W^+]$ and $Var[W^+]$ are obtained respectively from the model $m(Z)$ and from $1/(4\,(2\alpha(I)+1))$.

## 2.5 Estimation of scaling models

Table 1 summarizes the dependencies of each MRC model to either temporal scale, intensity and/or asymmetry and indicates the parameters to be estimated for the corresponding scaling models.

Whatever the MRC model, the estimation of the different parameters required for the scaling model for $p_x$ is independent of those related to $\alpha$. The estimation of the scaling sub-model for the probability asymmetry index $\varphi$ and for the mean $m$ of the distribution $f_{W^+}$ is also independent of the estimation of the scaling models for the dependency to intensity/temporal scale. In all cases, the estimation is also sequential: in a first step, the empirical value of the variable of interest (e.g. $p_x$, $\alpha$, $\varphi$ or $m$) is calculated for different classes of disaggregation configuration (e.g. different temporal scales, or different classes of intensity, or different classes of asymmetry index) and in a second step, the relevant scaling model is fitted to those empirical values by the method of least square errors.

This is illustrated in panels (b) and (c) of Figure 3 for the asymmetry scaling model $\varphi(Z)$ and $m(Z)$. The models of equations 16 and 17 are fitted (the plain black lines) on the empirical values of $\varphi$ and $m$ obtained for different asymmetry index classes (black dots). Empirical values of $\varphi$ and $m$ are calculated for each Z-index class by ignoring the intensity class and temporal scale. Estimation is also made on a seasonal basis. The estimation process for the parameters of the other scaling models is similar. It is described in detail for all models in Appendix A. An example of model fit is given in Figure 1a, c and d (plain lines) for the scaling models $p_x(I,\tau)$, $g(I)$ and $h(\tau)$ used in model A. For the scaling model $\alpha(I)$ used in model B, an example is given in Figure 1b.

**Table 1.** The 4 models compared and the parameters to be estimated for the scaling sub-models accounted for. The - symbol denotes a configuration where the scaling dependency is disregarded. The total number of parameters per season and station is given in column "Nb. of params".

| Model | Nb. of params | $p_x$ or ($p_{10}$ and $p_{01}$) | | | $f_{W^+}$ ($\alpha$ only or $\alpha_1$ and $\alpha_2$) | | |
|---|---|---|---|---|---|---|---|
| | | Intensity | Temporal scale | Asymmetry | Intensity | Temporal scale | Asymmetry |
| **A** | 9 | $p_x(I,\tau) \to a_\mu, b_\mu, a_\sigma, b_\sigma$ | | - | $\alpha(\tau, I) \to c_0, c_1, c_2, \alpha_0, H$ | | - |
| **A$^+$** | 11 | $p_x(I,\tau) \to a_\mu, b_\mu, a_\sigma, b_\sigma$ | | $\varphi(Z) \to \nu$ | $\alpha(\tau, I) \to c_0, c_1, c_2, \alpha_0, H$ | | $m(Z) \to \lambda$ |
| **B** | 3 | $p_x(I) \to \mu, \sigma$ | - | - | $\alpha(I) \to K$ | - | - |
| **B$^+$** | 5 | $p_x(I) \to \mu, \sigma$ | - | $\varphi(Z) \to \nu$ | $\alpha(I) \to K$ | - | $m(Z) \to \lambda$ |

Precipitation data aggregated at six temporal resolutions are considered for model estimation (40, 80, 160, 320, 640, and 1280 min). 40-minute precipitation data are obtained from the 10-minute time series available for the stations. The 40-minute time series are aggregated to 80-minute time series to calculate the observed breakdown coefficients $W$ using non-overlapping adjacent pairs of precipitation amounts for this cascade level. The aggregation procedure is repeated until the time series

reaches the temporal resolution of 1280 minutes. These different series are used for the estimation of the BDCs relative to each temporal scale and are also used as a reference for the evaluation of the disaggregation models (see Sect. 2.6).

Note that the 1280-minute aggregated temporal scale does not correspond to the daily resolution of 1440 minutes. Following Molnar and Burlando (2005), the first and last 80-minute records of each day were thus discarded from the initial observed times series before aggregations. The resulting time series, with truncated days of 21.3 hours, was used as a reference "daily"

time series for model estimation and evaluation.

As reported in previous works, the precipitation measurement resolution, defined by the precipitation tipping bucket of automatic stations, 0.1 mm here, is very likely to introduce artefacts and/or biases in different statistical characteristics of precipitation at fine temporal scales, especially sub-hourly ones and can impact the empirical distributions of the breakdown coefficients $W$ (e.g. Olsson, 1998; Rupp et al., 2009; Licznar et al., 2011; Paschalis et al., 2012).

Following previous works, the breakdown coefficients $W$ obtained from precipitation amounts below a given precipitation threshold were discarded for the present study. A threshold of 0.8 mm is applied for the estimation of $\alpha$ in all models, and for the estimation of $p_x$ in models B and B$^+$ (see Sect. 5.1 for further discussion on this issue).

## 2.6 Experimental setup

The 10-minute observational records aggregated to the resolution of 1280 minutes are disaggregated back to the 40-minute

resolution using models A, B, A$^+$ and B$^+$. Since models are stochastic, the disaggregation is performed 30 times for each model.

The performance of a given model is evaluated by its ability to reproduce standard statistical metrics of precipitation, such as the standard deviation of precipitation amounts, the probability of precipitation occurrence, and return levels of maximum

precipitation amounts for given return periods. In addition, the temporal autocorrelation, wet-dry transition probabilities and the duration of wet and dry spells are used to assess the ability of the models to reproduce the temporal persistence of precipitation. Evaluations are carried out on a seasonal basis for all available stations and for all temporal scales involved in the generation process (i.e. 40 to 640 minutes).

Different evaluation criteria are used. For most evaluation metrics, the variability between generated scenarios is small to very small. In this case, the evaluation criterion is the absolute error between the simulated and the observed metric, averaged over the different scenarios (and possibly temporal scales and seasons). To assess the performance of a given model across multiple sites, single-site performances are averaged over the different stations. We refer to this performance criterion as the Mean Absolute Error (MAE). For some metrics, the percentage absolute relative error is considered (dividing for each station, season, temporal scale, the absolute error by the observed value of the metric), giving the Mean Absolute Percentage Error (MAPE, Hyndman and Koehler, 2006).

For precipitation maxima, the variability between scenarios is often large. In this case, we apply the CASE evaluation framework proposed by Bennett et al. (2018) for which at-site model performances are categorized as "good", "fair" and "bad". For a single metric at a given site, 90% and 99.7% probability limits are obtained from the set of simulated metrics and are compared with the observed metric. Then, the performance is categorized as:

- "good", if the observed metric is inside 90% limits of simulated metrics,

- "fair", if the observed metric is outside of 90% limits of simulated metrics but within the 99.7% limits of simulated metrics or absolute relative difference between the observed metric and the average simulated metrics is 5% or less, and,

- "bad", otherwise.

To assess the performance of a given model across multiple sites, single-site performances are summarized as percentages of "good", "fair" and "bad" cases among sites and seasons. The variables evaluated here are the return levels estimated for 5- and 20-year return periods. The 5- and 20-year return levels, respectively, are estimated empirically in a Gumbel plot from a linear interpolation between the two observed (or simulated) annual maxima which have the empirical return periods the closest to 5 and 20 years, respectively. The empirical return period of each observed (or simulated) annual maximum is simply obtained from its empirical non-exceedance probability estimated with the Gringorten plotting position formula (Gringorten, 1963).

## 3   Application to Swiss data

The four models are applied to 81 stations of the Swiss meteorological observation network, a relatively dense network, with high-quality observational data (Figure 4). The stations have at least 20 years of data available through the period 1980-2020 and use tipping bucket rain gauges with a sampling resolution of 10 minutes and tip volume of 0.1 mm. Switzerland has a complex topography, with 60% of the territory covered by the Alps. Together with the Swiss Plateau and Jura mountains on the north, they are the main features of the landscape. The complex topography induces very different weather and precipitation regimes as described in the precipitation climatology for the European Alps of Frei and Schär (1998). For instance, the main

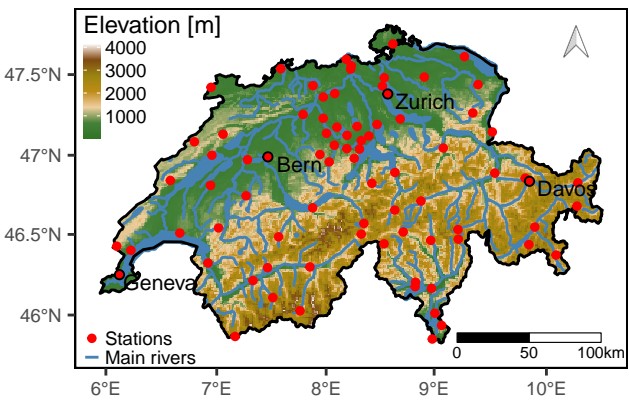

**Figure 4.** Map of Switzerland and gauge locations.

topographic slopes at the rim are responsible for enhanced precipitation and associated rain-shadowing of inner-Alpine sectors. The Swiss Plateau tends to be sheltered by the Jura mountains, resulting in less precipitation from northwest fronts during winter than the Jura mountains (Baeriswyl and Rebetez, 1997).

If large precipitation amounts can be observed during winter and spring due to long stratiform and orographic precipitation events, intense precipitation events are often observed in summer and fall due to the topographically triggered convective events (Frei and Schär, 1998). Mediterranean cyclonic activity in autumn brings heavy precipitation in the region of Ticino, in the south of Switzerland, making autumn the main precipitation season in this region (Molnar and Burlando, 2008).

## 4   Results

We first focus on results obtained for the target 40-minute temporal resolution, namely for a set of standard statistics (Sect. 4.1) and for 5- and 20-year return levels of annual maxima (Sect. 4.2). Next, in Sect. 4.3 we present how results vary for intermediate disaggregation temporal resolutions and for the different seasons.

### 4.1   Standard statistics

Simulated values obtained for a set of standard statistics at a resolution of 40 minutes are confronted to observed ones in Figure 5. For standard deviation and proportion of wet 40-minute time steps, results are satisfying whatever the model. Slightly better results (i.e. smaller MAE values) are obtained with models B and $B^+$ for standard deviation (Figure 5a) and with models A and $A^+$ for the proportion of wet steps (Figure 5b).

Differences between models are more important for statistics related to precipitation persistence and intermittency. The best-performing model depends on the statistic but, whatever the statistic, the performance of the model is always drastically improved when precipitation asymmetry is accounted for (see model $A^+$ vs model A, and model $B^+$ vs model B). At a 40-minute resolution, this is illustrated with results obtained for the wet/dry transition probability, lag-1 autocorrelation, and mean

duration of wet spells in panels (c), (d), and (e) of Figure 5 respectively. For all stations and all seasons, the large over- or under-estimation obtained with models A and B is largely reduced and even tends to disappear when asymmetry is accounted for (models $A^+$ and $B^+$). In the same manner, much better results are obtained for all other similar statistics and all other temporal scales, as indicated by the results obtained for lag-2 autocorrelation, mean duration of dry spell and all the other wet/dry transitions probabilities (dry/dry; dry/wet; wet/wet) provided in the Supplementary Material (SM) (see Figure S3, S4 and S5).

Some differences are also noticed depending on whether or not the dependency on the temporal scale is taken into account (i.e. model B vs model A, and model $B^+$ vs model $A^+$). While these differences are sometimes non-negligible, they are much smaller than the obtained differences depending on whether or not the dependency on the asymmetry is included. For example, at the 40-minute resolution, the observed autocorrelation coefficients are better reproduced with a dependency on the temporal scale (Figure 5d for lag-1 and Figure S3 for lag-2 in the SM). For durations of wet and dry spells and the different wet/dry transitions probabilities, the consideration of dependency on the temporal scale has a limited influence (see panels (c) and (f) of Figure 5 for the duration of wet spell and wet/dry transition probability, and Figure S4, S5 in the SM for other statistics).

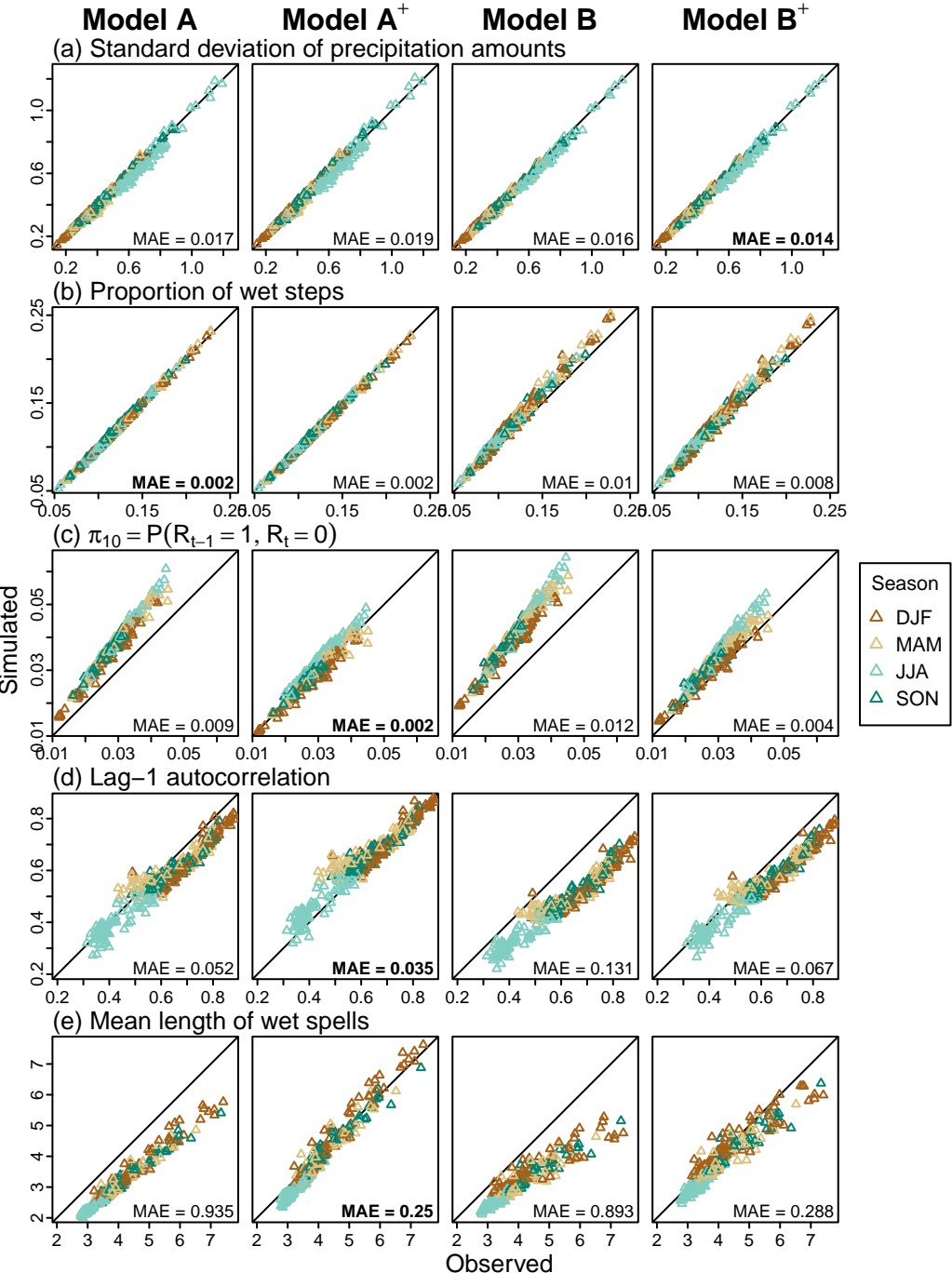

**Figure 5.** Observed versus simulated statistics for each considered model at a 40-minute temporal resolution for different metrics. Each triangle represents a site and a season. The triangles for the simulated metrics correspond to the median of the 30 statistics obtained from the corresponding 30 simulated scenarios. MAE values over all sites and seasons are indicated in the bottom-right corner, the lowest MAE obtained over the four models is indicated in bold. Pearson's autocorrelation coefficients are estimated using the function `acf` implemented in R, R Core Team (2022).

## 4.2 5- and 20-year return levels

The ability of the models to simulate relevant return levels for the 5- and 20-year return periods at the 40-minute temporal resolution is presented in Figure 6. The percentages of station/season configurations corresponding to the "good", "fair" and "poor" CASE criteria are summarized with the horizontal green/yellow/red bar, respectively, in each graph.

Results are very similar from one model to the other. The percentage of sites with "good" performance is slightly higher for models B and B$^+$ but MAPE values are very similar for all models. Contrary to most of the other considered metrics, the consideration of asymmetry has almost no influence here.

All models are able to reproduce the large variety of 5- and 20-year return levels observed in Switzerland between sites and seasons. They also all perform rather well for both return periods. For most sites and seasons, however, return levels are slightly underestimated. At the 40-minute resolution, the underestimation is stronger in spring and summer for some sites (Figure 7). This point will be further discussed in Sect. 5.5.

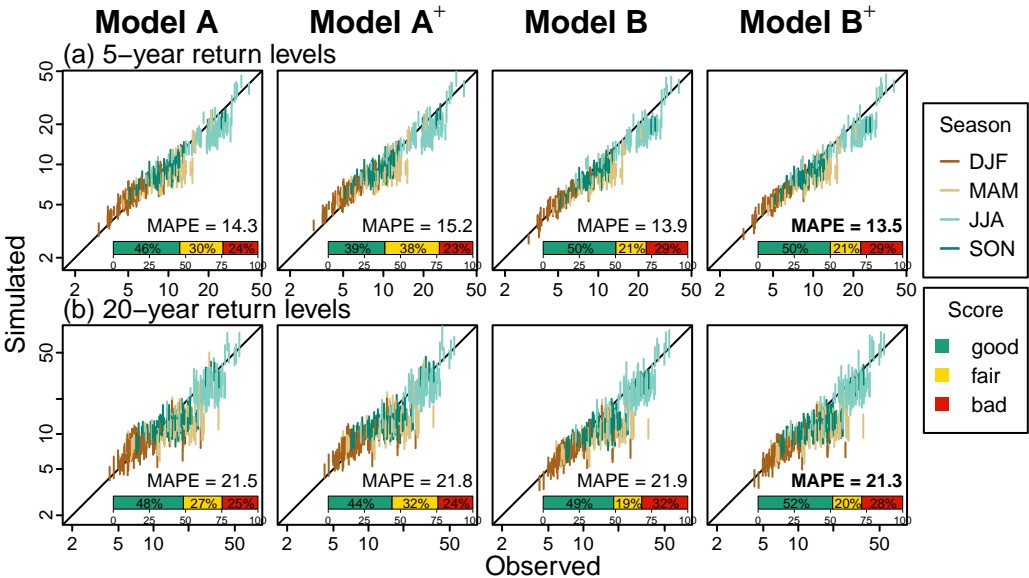

**Figure 6.** Observed versus simulated return levels at the 40-minute temporal resolution for (a) 5-year and (b) 20-year return periods, for each model and at each site. Vertical bands indicate the 90% CI limits obtained from the 30 simulated time series for each station and each season (one colour by season). Horizontal coloured bars indicate the percentage of "good", "fair", and "poor" performance as assessed by the CASE framework for all sites and seasons (81 × 4 = 324 cases). The return levels are estimated empirically using the Gringorten plotting position formula (Gringorten, 1963). MAPE values over all sites and seasons are indicated in the bottom-right corner, the lowest MAPE obtained over the four models is indicated in bold.

### 4.3 Intermediate temporal scales and dependency to seasons

Figure 7 presents the performance of each model for the reproduction of the lag-1 autocorrelation, mean length of wet spells, 5-year and 20-year return levels at different temporal resolutions. Clearly, for almost all statistics, the performances depend on the temporal resolution. For the intermediate disaggregation resolutions, the best-performing models are not necessarily the same as those found at a 40-minute resolution. For instance, between models A and B, in regard to the reproduction of the lag-1 autocorrelation, the best model at a 40-minute resolution is model A, while model B is the best model at 160-minute and coarser resolutions. For the mean length of wet (and dry) spells and for 5- and 20-year return levels, models A and B provide similar performance for 40-minute although model B tends to perform better at 160 minutes and at coarser resolutions.

At 40-minute resolution, accounting for asymmetry significantly improves the reproduction of all statistics related to precipitation persistence and intermittency, but does not influence the reproduction of standard deviation, wet step probability, and return levels. Similar results are obtained for the intermediate temporal resolutions.

As already mentioned previously, results can also depend on the season. Results for spring and summer are often similar and contrast with the results obtained in winter and autumn (see Figure 7).

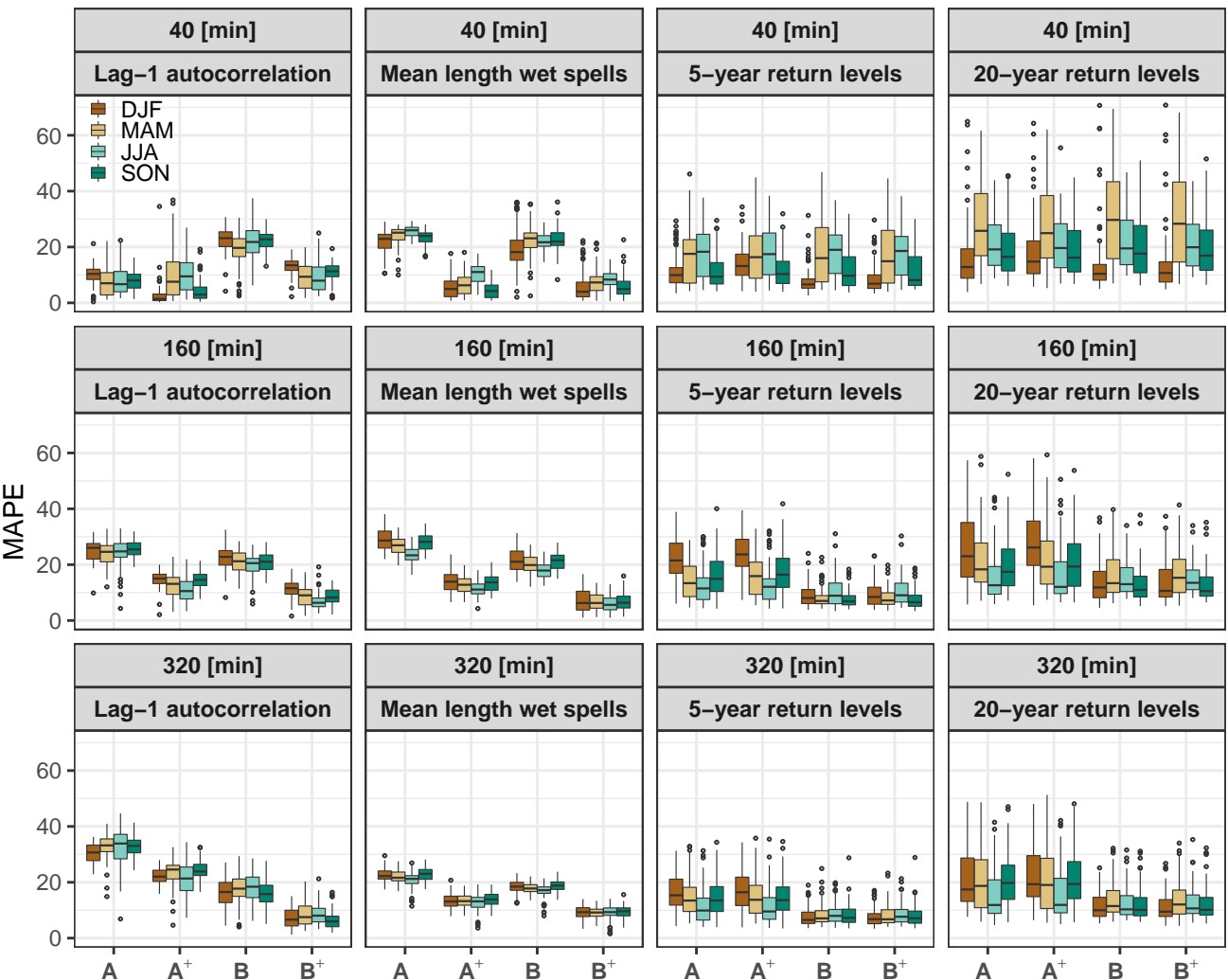

**Figure 7.** Mean Absolute Percentage Error (MAPE) as a function of the temporal aggregation level and season for lag-1 autocorrelation, mean length of wet spells, 5-year and 20-year return levels. Each boxplot summarizes the single site performances obtained for the 81 stations.

## 5 Discussion

### 5.1 Scaling MRC parameters with temporal scales

As highlighted in many previous works, the distribution of the breakdown coefficients $W^+$ depends on the temporal scale, precipitation intensity, precipitation asymmetry and possibly other factors. Even though it was done in a discrete way by conditioning MRC parameters on a few external pattern classes, the dependency to the precipitation asymmetry has been widely accounted for in empirical approaches. However, to our knowledge, except for the pattern-based MRC model presented in Hingray and Ben Haha (2005), all analytical MRC developments have disregarded this dependency, most of them focusing on temporal scale and intensity dependencies.

According to the results shown in Sect. 4, the dependence on the temporal scale and its added value are not as pronounced as usually considered. In the present study, model A, which is the model considered by Rupp et al. (2009), and later by Paschalis et al. (2014), includes the dependency on the temporal scale and precipitation intensity. Model B, a simplification of model A, disregards temporal scale dependency. Besides the fact that it increases the number of parameters to be estimated (six additional parameters), model A is found to have similar performances to model B and even worse for coarse temporal scales. Note that Molnar and Burlando (2005) and Rupp et al. (2009) also show this limited advantage of the temporal scale dependency.

Two issues must be put in perspective with the preceding statements. The first one is related to the recording precision, which induces a relatively high frequency of $W = 1/2, 1/3, 2/3$, and to a lesser extent of $W = 1/4$ and $3/4$. This precision artefact is actually detrimental for a relevant estimation of the cascade generator $\Gamma$ characteristics for small temporal scales and low intensities (not for moderate to high intensities, see, e.g. Olsson, 1998; Rupp et al., 2009; Paschalis et al., 2012). Rupp et al. (2009) suggested that some of the dependency on intensity was the result of this artefact precision. We argue that a large part of the dependency on the temporal scale is also a result of this precision artefact. This is strongly suggested here concerning the probability $p_x(I)$. As illustrated in Figure 1a, the $p_x(I)$ relationship for the Zurich station differs from one temporal scale to the other. The difference between the curves, which is mainly observed for small to very small intensities, depends a lot on the low precipitation threshold considered for selecting precipitation data (not shown). The differences between the curves almost disappear when precipitation amounts smaller than 0.8 mm are removed from the analysis (see Figure S2 in the SM for the Zurich station). Without the small precipitation amounts, the scaling model for $p_x(I)$ appears to not depend on the temporal scale anymore and motivates the simplification of model B considered here. Further investigations will be worth to assess whether this holds true in other regions. This issue also calls for the development of a robust assessment framework (to be defined) that would be able to disentangle, if existent, the genuine dependence on the temporal scale from the one induced by the precision artefact.

Note that these conclusions were also obtained in auxiliary analyses we carried out on jittered high-resolution precipitation data (not shown). Random perturbations were added to the original observed 10-minute time series before model estimation procedure as in Licznar et al. (2011). In our analysis, the jittering process was designed as an attempt to mimic the measurement process of the tipping bucket rain gauge. In a tipping bucket with 0.1 mm precision device, one rainfall pulse is recorded within a given 10-minute time step once the 0.1 mm rainfall bucket is filled, regardless of the duration required to fill the bucket. In

reality, the recorded 0.1 mm measurement is likely to partially belong to the current time step and partially to the previous time steps. The jittering process reallocates the first recorded 0.1 mm of each time step to the current and previous time steps. This process results on the elimination of rounded quantities of precipitations (k x 0.1 mm; k = 1, 2, ...n), especially small ones (0.1, 0.2, 0.3, ... mm), and consequently on the removal of over-represented values of BDCs, $W = 1/2, 1/3, 2/3, 1/4$ and $3/4$. This leads in turn part of the "scaling dependencies" mentioned above to disappear.

Another issue concerns the parameter $\alpha$ of the distribution $f_{W+}$. For this parameter, the dependence on the temporal scale is poorly understood. For Zurich, rather large deviations are obtained between modelled and empirical estimates whether this dependence is taken into account (model A, see scaled parameter $g(I)$ in Figure 1d) or not (model B, see Figure 1b). Deviations are mainly observed for large intensities in model A and for moderate intensities in model B. The temporal scale dependency assumption also seems to be detrimental for the disaggregation of large precipitation amounts. In all cases, none of the two models A and B are fully satisfying for representing $\alpha$, similar results being obtained for all other stations considered in this work. The dependency on the temporal scale needs to be better characterized in future works, along with alternative scaling models.

## 5.2 Scaling MRC parameters with precipitation asymmetry

The statistics related to precipitation persistence have always been found to be significantly underestimated by analytical MRC models (e.g. Hingray and Ben Haha, 2005; Rupp et al., 2009; Paschalis et al., 2012). Paschalis et al. (2014) actually suggest that analytical MRCs, by construction based on a symmetric cascade generator, are unable to account for the multiscale autocorrelation structure of precipitation. The results shown in Hingray and Ben Haha (2005) indicate that this issue could be at least partly fixed when the precipitation asymmetry is accounted for in the model. These insights are thus confirmed with the present work, for a large variety of climate contexts found in Switzerland and for all seasons.

The dependence on precipitation asymmetry, acknowledged for a long time, was never accounted for in a fully analytical MRC framework. This study fills this methodological gap. We show that the dependency on the asymmetry can be modelled with a two-part analytical scaling sub-model, where the dry probability asymmetry ratio $\varphi$ and the mean $m$ of the positive BDCs distribution $f_{W+}$ depend on the precipitation asymmetry index $Z$. For Swiss stations, the dependence of the cascade generator properties on precipitation asymmetry seems to be much larger than that on temporal scales. The added value of introducing precipitation asymmetry in the model is clear: all statistics relative to precipitation persistence are much better reproduced, whatever the station, the season, and the temporal scale.

Accounting for precipitation asymmetry seems to be of crucial importance to achieve the generation of a time series with relevant properties. This is also suggested by the results of some auxiliary evaluations described below. As shown previously in Section 2.3, the statistical distribution of $W$ is expected to strongly depend on the external pattern of precipitation and more precisely on the value of the asymmetry index $Z$. When the asymmetry index $Z$ is larger, the probability $p_{01}$ (resp. $p_{10}$) is expected to be smaller (resp. larger) and the mean of the distribution is expected to be larger (cf. ECDFs of $W$ for Zurich data in Figure 3).

Figures S12 and S13 in the SM demonstrate that the dependency of ECDF of BDCs on $Z$ is almost perfectly reproduced with models $A^+$ and $B^+$ for Zurich data. Conversely, when the cascade generator does not account for precipitation asymmetry, as is the case with models A and B, the dependence of ECDF of $W$ on $Z$ is no longer reproduced, which is not surprising due to the inherent symmetric formulation of cascade generators.

Besides, the statistics of a precipitation time series, e.g. the standard statistics and the ECDF of $W$, are not expected to
change when the time series is offset in time by a small time duration. This specific point is investigated here with an offset experiment similar to that presented by Rupp et al. (2009). Four 40-minute time series are derived from the initial 10-minute time series by aggregating 10-minute data using four different time offsets: no offset, 10, 20, and 30-minute. When the offset experiment is applied to observed data, the considered statistics, estimated on the four time series are similar. Thus, the expected offset-independence property is satisfied as shown in Figure S10 for standard statistics of 12 stations and in figures S12 and S13
for the ECDF of $W$ for Zurich data. When the offset experiment is applied on times series generated with the disaggregation models, the offset-independence property is not always satisfied. For illustration, the disaggregated 40-minute time series were further disaggregated to 10-minute time series (for the 20-minute and 10-minute temporal scales, the parameters of the cascade generator are obtained with the scaling models). For each 10-minute time series scenario, four offset 40-minute time series were produced with four different offsets as for the observations. In general, the statistics obtained with disaggregated scenarios from
models A and B (without asymmetry) are much more sensitive to temporal offset than those obtained from models $A^+$ and $B^+$. For models A and B, the estimates obtained for the three non-zero offsets (10, 20, 30-minute) are often significantly different from the reference estimate (with the 0-minute offset). This is the case for standard statistics as highlighted in Figure S11 and it is even more accentuated for the ECDFs of $W$ as illustrated in Figures S12 and S13 in the SM. Including the asymmetry in the generator makes the ECDFs rather insensitive to the offset.

As mentioned previously, the asymmetry of the cascade generator is mainly disregarded in analytical MRCs but it is considered for a long time in empirical ones by estimation of the cascade generator for different external patterns classes. Conditioning the parametrisation of any analytical model on external patterns classes is also possible. Is our continuous asymmetry approach of interest when compared to a class-conditioned approach? To address this question, we considered two more models "A position" and "B position" ("Ap" and "Bp"). These two models are based on models A and B respectively, but are estimated by
conditioning the estimation on four external pattern classes, i.e. the four position classes (starting, enclosed, ending, isolated) considered in McIntyre et al. (2016). Again, the observed quasi-daily amounts were disaggregated to time series of 40-minute resolution with all models. As shown in figures S14 and S15 in the SM, results depend on the considered statistics. Overall in our case, conditioning the analytical models to the position class improves the performance of classical unconditioned analytical models; nevertheless, it is less efficient than considering scaling laws with a continuous asymmetry index. These
interesting results will be worth further investigation, especially in other climates. We believe that the better performance of models $A^+$/$B^+$ over models Ap/Bp is due to the fact that they are additionally able to make a distinction between different starting sequences (or different ending sequences) as the asymmetry index is also a measure of the intensity of the asymmetry (i.e. steep decreasing intensity over the three consecutive precipitation amounts or only slow decreasing). As they are additionally highly parsimonious, models $A^+$ and $B^+$ appear to be promising alternatives to the class-conditioned models.

## 5.3 Spatial variability of parameters and the potential for a regional MRC model

In order to apply a MRC model in locations where only daily data are available, the possibility to develop a robust regional model for model parameters is necessary. The smaller the number of parameters, the more robust and easier the regionalisation is expected to be. In this regard, model $B^+$ (5 parameters) is likely to be much more appropriate for regionalisation than model $A^+$ (11 parameters).

Another important factor that can jeopardize the success of parameter regionalisation is the spatial variability of parameters. In case of no evident relationship to some geographical features (e.g. topography), smooth spatial variations of parameters often help achieve robust regionalisation. The spatial variability of the five parameters of model $B^+$ is shown in Figure 8 where maps of estimates for $\mu, \sigma, K, \nu$ and $\lambda$ are presented for each season. Regardless of the parameters and throughout all seasons, the regional coherency is rather impressive. All parameters present significant spatial variability, reflecting the large variety of regional climates in Switzerland, nevertheless, the spatial variations are very smooth at sub-regional scales and from one region to the other. These results give increased confidence in the relevance of estimates and in the robustness of the model. They also suggest that a relevant regional model will be possible for all parameters, enabling the temporal disaggregation of coarse-resolution precipitation data anywhere in Switzerland. The performance of such a regional MRC model will be further investigated in future works.

## 5.4 Seasonal variability of parameters

Additionally, the maps presented in Figure 8 depict the seasonal variations of the various parameters. As already shown in previous works (e.g. Molnar and Burlando, 2005), the season is actually a much larger factor of variability than the location. The contrast between winter and summer is particularly marked. Whatever the parameter, the values for spring and autumn are similar to each other. This seasonality reflects the different types of rainfall events throughout the year, with more stratiform events in winter, more convective ones in summer, and mixed types in intermediate seasons. Stratiform events are generally persistent in time and with a large spatial extent, contrary to convective events which are often localized and with a short duration. The partition of any precipitation amount in two parts is thus expected to be smoother in winter, with more frequent partitions of similar rainfall amounts on the two subdivisions, and more dependency on the external precipitation pattern. This is reflected by higher $p_x$ values, higher $\alpha$ values as well as higher sensitivity to precipitation asymmetry. This is also shown by the values of all scaling parameters in winter: lower values of $\mu$ and $\sigma$ for the intermittency probability scaling sub-model, higher value of $K$ for the no-dry subdivision scaling sub-model, lower value of $\nu$ and higher value of $\lambda$ for the asymmetry scaling sub-models. As suggested by Rupp et al. (2009), an alternative to the seasonal stratification could consist of a parameterization conditional on the type of precipitation events, especially for intermediate seasons where different types of events are observed.

These results suggest the importance of seasonal stratification for model parameter estimation. The interest of seasonal stratification is even more obvious when simulations are carried with a model where parameter dependency on season was ignored. This is illustrated in the figures S8 and S9 in the SM where the seasonal performance of all models are presented for

a configuration where one single set of parameters has been estimated for all seasons together. The performance of the models without seasonal stratification degrades a lot for a number of statistics, especially in winter and summer seasons.

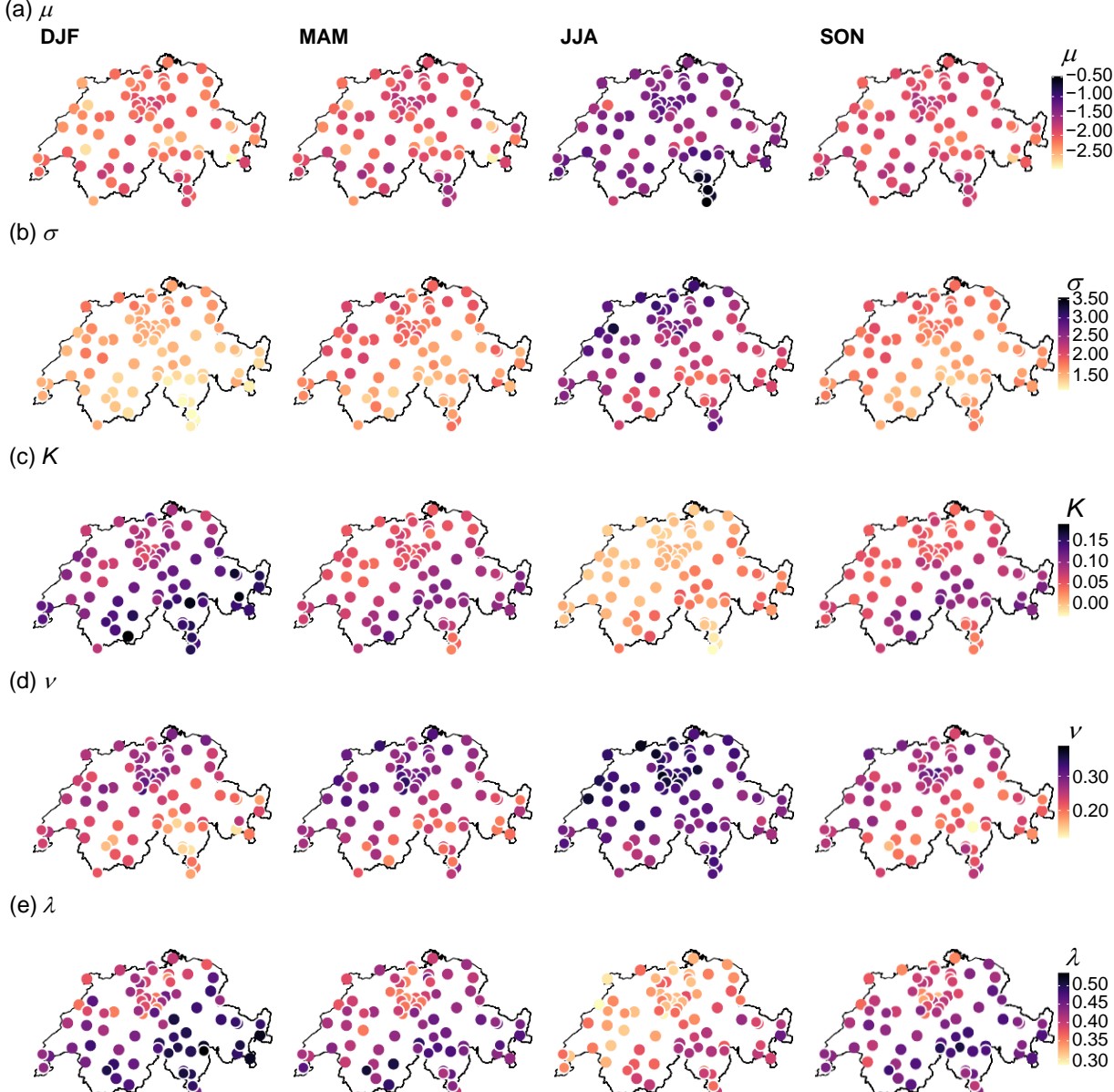

**Figure 8.** Maps of Switzerland showing spatial and seasonal variations of the estimated parameters for model B$^+$. Each row corresponds to one parameter of the model. Parameters representing the relationship between intermittency and intensity are those of the equation (9): (a) $\mu$ and (b) $\sigma$. (c) Parameter $K$, equation (13), related to the dependency of the distribution of positive weights, $f_{W+}$, on the precipitation intensity and finally the parameters $\nu$ and $\lambda$ in (d) and (e), respectively, hold the dependency of the cascade weights on the precipitation asymmetry through equation (16) and equation (17). The colour of the circle indicates the value of the parameter estimate for the respective station and season.

## 5.5 Asymptotic assumption of no-dry subdivisions probability for high intensities

An opportunity for model improvement concerns the scaling sub-model for the no-dry subdivisions probability $p_x$. As shown in Sect. 4.2, all the models tend to underestimate spring and summer extreme events on some sites. Figure 9a shows that, for model B, this principally occurs in northern Switzerland for spring and summer seasons. Very similar maps are obtained for the other models (not shown). A plausible explanation is that the model used for $p_x$ is not valid for very large intensities.

Following previous works, $p_x$ is assumed to be related to precipitation intensity via the erf function of equation (6). By construction, for large precipitation intensities, this scaling model has an asymptotic value of one, and $p_{01}$ and $p_{10}$ tend to zero. In other words, very large precipitation amounts are systematically divided into two non-zero parts. However, the probability $p_x$ has no reason to tend to 1, especially at coarse temporal scales (see illustration in Figure S2 in the SM). For instance, large daily precipitation amounts in summer can result from intense convective events with short duration (less than an hour). In this case, the daily amount must not be subdivided nor spread over multiple high-resolution time steps. The BDCs for such events are either 0/1 or 1/0 and the probability to have the 1/1 configuration is expected to be possible for high resolutions only. When $p_x$ is overestimated for large intensities, the 1/1 configurations (no-dry subdivisions) are thus over-represented and lead to an underestimation of extreme values.

The misrepresentation of $p_x$ for large intensities is illustrated with the mean deviation obtained between modelled and empirical $p_x$ values for intensities larger than 1 mm h$^{-1}$ in Figure 9b. Whatever the season, the spatial pattern of these deviations fits very well with that obtained for the underestimation of the 5-year return level shown in Figure 9a. The larger the deviation, the worse the underestimation. Relaxing the current asymptotic assumption for the $p_x$ model may improve the simulation of extremes.

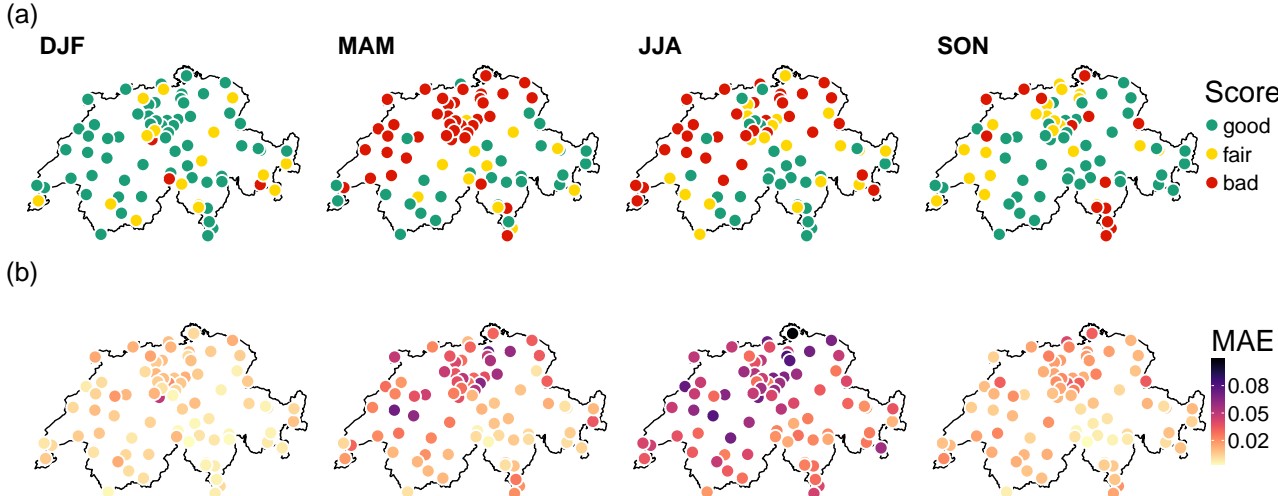

**Figure 9.** (a) Scores as assigned by CASE framework for the reproduction of 5-year return levels at 40-minute temporal resolution by model B for each site and each season. (b) Mean absolute errors (MAE) between observed and modelled $p_x$ values for precipitation intensities higher than 1 mm h$^{-1}$ with model B, see equation (9).

## 6 Conclusions

According to many previous works, the distribution of the breakdown coefficients (BDCs) used in MRC models depends on a number of factors, including the temporal scale, precipitation intensity and the external pattern of the local precipitation sequence.

In the present study, we compare different MRC models, with/without the scaling dependency to the temporal scales, and with/without the scaling dependency to the external pattern of precipitation. Conversely to the scaling dependency to precipitation intensity and asymmetry, the scaling dependency to temporal scales is not obvious and its added value in terms of model performance is less clear than what was suggested in previous works. Moreover, accounting for a dependency on the temporal scales drastically increases the number of parameters to be estimated (six more in the present case), which is especially expected to make the model much less robust and less appropriate for further regionalisation works.

The dependency on the external pattern of precipitation is shown to be important. In previous studies, it was mainly accounted for with empirical models where the BDC distribution was conditioned on different external pattern classes. To our knowledge, although determinant, it was never accounted for in an analytical scaling framework, which also accounts for temporal scale and intensity dependencies. Our work presents a unified analytical MRC modelling framework that allows the cascade generator to depend in a continuous way on the temporal scale, precipitation intensity, and precipitation asymmetry. The continuous dependency of the cascade generator on the precipitation asymmetry index, which is introduced here, allows for the interpretation of the presented asymmetry sub-model as an extension of the position-dependency approach already considered in several previous works. This sub-model could be easily assimilated in other multiplicative cascade models, either micro-canonical or canonical ones.

Initially, we demonstrate the feasibility of characterizing the external precipitation pattern with a hidden BDC, the so-called precipitation asymmetry index. We show that the larger the deviation of this index from $0.5$ (the index value for a symmetrical precipitation configuration), the larger the asymmetry of the distribution of the BDCs. The relationships with this asymmetry index are modelled with two scaling sub-models. The first sub-model represents the dry probability asymmetry ratio $\varphi$ that quantifies the asymmetry between $p_{10}$ and $p_{01}$, i.e. the probabilities that all the precipitation amount is attributed exclusively either to the first or to the second subdivision, respectively. The second sub-model is related to the mean $m$ of the distribution of positive BDCs, $f_{W^+}$, which is equal to $0.5$ if the asymmetry is disregarded.

Accounting for precipitation asymmetry in the cascade generator preserves the good performances of MRCs concerning statistics related to precipitation distribution (standard deviations and precipitation extremes) and improves other aspects of the disaggregated precipitation time series. For the Swiss context considered in our work:

- Accounting for precipitation asymmetry leads to significant model improvements for all statistics related to the temporal persistence and intermittency of precipitation, which are known to be difficult to simulate with standard MRC models.

- The statistical distribution of BDCs is expected to strongly depend on the external pattern of precipitation. This dependency is well (resp. is not) reproduced when precipitation asymmetry is included (resp. not included) in the MRC.

- The statistics of a precipitation time series are not expected to change when the time series is offset in time by a small time duration. This offset-independence property is well (resp. is not) satisfied when precipitation asymmetry is included (resp. not included) in the MRC.

Among the four different MRC models considered here, the one that accounts only for precipitation intensity and asymmetry seems promising. It performs very well for all considered statistics, for all seasons and for all temporal resolutions. It is, moreover, very parsimonious, with only five parameters. The five parameters are almost all independent from each other and can be estimated in a robust way, which avoids equifinality issues (Beven, 2006; Bárdossy, 2007). Whatever the parameter, the regional coherency is rather impressive. While all parameters present significant spatial variability reflecting the large variety of regional climates in Switzerland, the spatial variations of parameter estimates are very smooth. This suggests that a relevant regional model will be possible for all parameters, allowing in turn the temporal disaggregation of coarse-resolution precipitation data anywhere in Switzerland. This disaggregation modelling framework is promising and could be also suited for other climate contexts worldwide. Additional applications could help to better characterize its performance, its limitations, and its potential for regionalisation in such other contexts.

*Code and data availability.* The precipitation data used in this study are maintained by Swiss Federal Office of Meteorology and Climatology, MeteoSwiss (MeteoSwiss, 2021). It is available upon request at https://gate.meteoswiss.ch/idaweb/more.do (last accessed 10 December 2022).

The open-source code with routines allowing fitting and disaggregation of precipitation data based on the four MRC models presented in this study is available as an R package. It can be installed via GitHub: https://github.com/KaltrinaMaloku/disaggMRC.

*Author contributions.* Fund acquisition: BH; Data acquisition: KM; Experimental design: KM, BH, GE; Script development: KM, GE; model calibration, simulations and analyses: KM; Figures preparation: KM; Manuscript redaction: KM, BH, GE.

*Competing interests.* The authors declare no competing interests.

*Acknowledgements.* This research is part of the PhD thesis of Kaltrina Maloku. It has been supported by the Swiss Confederation, namely the Bundesamt fur Energie (grant no. SI/502150-01) and the Bundesamt fur Umwelt (grant no. SI/502150-01), through the project EXCH "Extreme floods in Switzerland". We thank the editor, Hannes Müller-Thomy and one anonymous reviewers for their constructive comments that helped us to improve the quality of this manuscript.

## Appendix A: Estimation of scaling model parameters

The parameters of the scaling models of each MRC model are estimated as follows:

- **Model A:** $p_x(I, \tau)$ is first estimated for different intensities classes $I$ and temporal scales $\tau$ as the proportion of wet time steps where both subdivisions are wet (coloured dots in Figure 1a). These estimates are used to fit the relationships (7) for $\mu(\tau)$ and (8) for $\sigma(\tau)$ using the method of least square errors, which leads to estimates of the parameters $a_\mu$, $b_\mu$, $a_\sigma$ and $b_\sigma$. For $\alpha$, it is first estimated for different classes of intensity $I$ and different temporal scales $\tau$ (coloured dots in Figure 1b) from the variance of the corresponding set of $W^+$'s by fitting a Beta distribution to the $W^+$ values with the method of moments (see equation (5)). It is also estimated for different classes of temporal scales $\tau$ (all intensities included; coloured dots in Figure 1c). The last estimates are used to fit the relationship (12) for $h(\tau)$ which leads to estimates of the parameters $\alpha_0$ and $H$. In the third step, the ratio $\alpha(I, \tau)/h(\tau)$ is estimated for different classes of intensity and different temporal scales (coloured dots in Figure 1d). These estimates are used to fit the scaling model $g(I)$ of equation (13) which lead to estimates of the parameters $c_0$, $c_1$, $c_2$. An example of model fit for $p_x(I, \tau)$, for $g(I)$ and for $h(\tau)$, is given in Figure 1a, c and d respectively (plain lines).

- **Model B:** The estimation of model B follows a similar but more direct sequential process than for model A. In model B, $p_x$ and $\alpha$ are assumed to only depend on the intensity $I$. In a first step, estimates of $p_x$ (resp. $\alpha$) are thus simply obtained for different classes of intensity (the empirical values of the W's calculated for all temporal scales are merged for the estimation). These $p_x$ (resp $\alpha$) estimates are then used to fit the scaling relationship (9) for $p_x(I)$ (resp. 13 for $\alpha(I)$) and obtain estimates of $\mu$ and $\sigma$ (resp. $K$). An illustration is given in Figure 1b for $\alpha$.

- **Model A$^+$/B$^+$:** Estimating the additional parameters related to the asymmetry follows an independent estimation process. Different $\varphi$ and $m$ estimates are first obtained for different Z-index classes (all intensities and all time scales are merged for the estimation). For $\varphi$, these estimates are obtained from equation (15), while estimates of $m$ are obtained as the mean of the $W^+$ values for different Z-index classes. The scaling models (16) for $\varphi(Z)$ and (17) for $m(Z)$ are then fitted on these estimates, which leads to estimates of the parameters $\nu$ and $\lambda$, respectively. An example of fit is illustrated in Figure 3b-c. The different steps necessary to obtain $p_{01}$, and $p_{01}$, and the two parameters $\alpha_1$, $\alpha_2$ of the asymmetric Beta distribution are described in Sect. 2.4.

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
