# Peer review of "Accounting for Precipitation Asymmetry in a Multiplicative Random Cascades Disaggregation Model"

_EGUsphere, 2023_

## Author Comment (AC1)

**Response to anonymous referee #1**

*We thank referee #1 for reading the manuscript carefully and providing thoughtful and constructive comments. The comments are noted with **RC**, our responses with **AC**, and the intended additions or changes in the manuscript are underlined.*

**RC1.1** Based on some exploratory analysis in Rupp et al. (2009), I do have a lingering doubt related to the apparent asymmetry and the need to explicitly account for it fully.

Asymmetry has been considered before although differently. For example, Olsson (1998) and Güntner et al. (2001) developed the distributions of the breakdown coefficients separately for time intervals that start or end a rainfall sequence or are within a rainfall sequence. They showed that starting/ending intervals had distinctly asymmetric distributions.

> **AC1.1** *We thank Reviewer #1 for this important comment. Yes, the dependency of the cascade generator on the external pattern of precipitation (i.e. the dependency of the cumulative distribution function (CDF) of the breakdown coefficients (BDCs) on the external pattern of precipitation), has been highlighted/commented in a number of publications. We were also aware of the work of Rupp et al. (2009) and of their analyses/doubts on the need/way to account for it in a MRC model.*
>
> *To our knowledge, accounting for this dependency in a MRC has mainly (almost only) been carried out with empirical MRC where the CDFs of the BDCs is estimated empirically and described with empirical CDFs. This allows to describe CDFs with different shapes. Analytical scaling MRC presented up to now, such as those described in Rupp et al. (2009), can conversely not account for this dependency. The reason is that the CDF of the BDCs is modelled with a symmetric distribution where one assumes equal wet/dry or dry/wet probabilities (p01 and p10) and a symmetric distribution of non-zero weights. For instance, the analytical model uses a symmetric beta distribution in Rupp et al. (2009), a mixed beta – normal distribution in Licznar et al. (2011a), a 3N-B distribution in Licznar et al. (2011b) where a symmetric beta distribution is combined to 3 two-side truncated normal distributions.*
>
> *One noticeable exception is the analytical MRC of McIntyre et al. (2016), who used a 2-parameters beta distribution. This distribution, which can be asymmetric, was used to model the asymmetric CDFs found for precipitation amounts in asymmetric precipitation sequences, namely the starting/ending intervals (for the respectively named "followed" and "preceded" precipitation amounts in their work). McIntyre et al. (2016) did not consider scaling models to estimate the CDF parameters (the CDF of the BDCs was estimated for each temporal scale for different categories of rainfall, defined from the volume of precipitation and the external pattern class (isolated, enclosed, preceded and followed)). The number of parameters for their model is then considerable, and is potentially too large to allow for a robust estimation.*
>
> *In our work, we fill this methodological gap. We consider an analytical model which is by construction asymmetric and whose parameters are related – thanks to simple scaling laws, to the asymmetry of the local precipitation pattern. This allows us to*

*keep the number of parameters very low, and to combine 1) scaling relationships with temporal scales and intensity and 2) "scaling dependency" on the external pattern structure. This is a first novelty. Next, we introduce the asymmetry index and we show that the asymmetry level of the cascade generator (i.e. the asymmetry level of the CDF used for the distribution of W's) depends on this index in a continuous way. This allows conditioning the MRC on the external pattern but without the need to consider external pattern classes. This is another strength of our approach.*

*As shown in Fig. 1, the "intensity" of the asymmetry of the local precipitation sequence can vary a lot from one external pattern to the other. This is not only a question of asymmetry class ('ending/starting", ...). As shown in Hingray and Ben Haha (2005) (Fig. 4 of this work given below), the distribution of W is not expected to be the same in configurations b) (high gradient in a steep descending pattern) and c) (low gradient in a steep descending pattern). The asymmetry index introduced in our work is shown to allow the distinction between such configurations.*

*The introduction of our paper will be strengthened to better describe previous issues.*

[Figure]

Fig. 4. Graphical illustration of the rainfall fraction $x_{OD}$ used in the pattern-based Microcanonical Random Cascade model (pbRC) for three rainfall sequences $[R_{i-1}, R_i, R_{i+1}]$: (a) left valley pattern; (b) descending pattern-high gradient; (c) descending pattern-low gradient. The grey rainfall volume spanning the three first 10-min bars and calculated according to Eq. (7) is $R_1 = G(0.5 \ast T) = x_{OD} \cdot R_i$. The white volume spanning the three last bars is $R_2 = (1 - x_{OD})R_i$.

*Figure 1. Illustration of possible breaking coefficient for a steep and a non-steep decreasing pattern precipitation sequence (from Hingray and Ben Haha (2005)).*

**RC1.2** Rupp et al. (2009) showed that models that did not explicitly incorporate asymmetry did not generate asymmetry at the time steps at which the rainfall was simulated. However, when they resampled their synthetic rainfall at an interval of the same duration but offset in time by small amount, asymmetry in the breakdown coefficients was introduced. The breakdown coefficients from the resampled series were remarkably like the breakdown coefficients from the observed data (see their Figure 16).

Rupp et al. (2009) concluded that at least some of apparent asymmetry in the breakdown coefficients arises from imposing a discrete, regularly timed sampling interval to an

irregularly timed phenomenon. To what degree, then, are the authors simply reproducing an artifact of sampling by incorporating asymmetry explicitly into their models? I think this issue needs discussion.

> **AC1.2** *We thank Reviewer #1 for this very interesting comment. We had indeed seen this analysis of Rupp et al. (2009) and their offset experiment. Note that they do not strictly conclude that at least some of the apparent asymmetry in the breakdown coefficients arises from imposing a discrete, regularly timed sampling interval to an irregularly timed phenomenon.*
> *They only say: "We suspect that the asymmetry in the starting and ending distributions is largely an artifact of sampling a semi-continuous and irregularly times process at discrete, regularly spaced intervals" (paragraph [41]).*
>
> *And after their "offset experiment", which results they presented in Figure 16. Their conclusion was (paragraph [44]):*
>
> *"While we do not present definitive evidence that the variability in the cascade weights among class intervals is completely an artifact of the sampling method, our preliminary analysis raises interesting issues that warrant further investigation. For one, if the dependency is largely an artifact, is the approach of Olsson [1998] and Gunter et al [2001] to reproduce it explicitly warranted, particularly because it substantially increases the number of model parameters required? Also, if we sample our rainfall events such that the sampling intervals begin and end exactly when the rain actually begins and ends, will weights in the middle of an event still differ from those near its onset or termination?"*
>
> *To our knowledge, unfortunately, no other work has been carried out to investigate these interesting issues. Our work strongly suggests that their "suspicion" was likely wrong. Our work suggests that there is no one single cascade generator for a given time scale and given intensity class, but a large variety depending on the asymmetry importance of the local precipitation pattern. The cascade generator is asymmetric and as it was demonstrated empirically by the works of Olsson (1998), Gunter et al. (2001), McIntyre et al. (2016) and others, this asymmetry is determined by the asymmetry of the local precipitation sequence around the precipitation amount to disaggregate. In line with the work of Hingray and Ben Haha (2005), our work additionally shows that the asymmetry of the cascade generator can be more or less important, depending the importance of the asymmetry of the local precipitation sequence.*
>
> *The main argument of Rupp et al. (2009) to their conclusion recalled above is based on their offset experiment. However, another conclusion could (should) likely be given. This is at least what suggests the following offset experiment we carried out to answer this issue raised by Reviewer #1.*
>
> ***Offset experiment.*** *The offset experiment was carried out on 40min time series data, but similar results are expected for other temporal aggregation levels. Precipitation data available for this experiment have a 10-minute resolution*

- *For a given station, in order to obtain 40-minute time series we aggregate 10-minute time series by using different time offsets: no offset, 10 min, 20 min and 30 min. Four time series have been thus obtained with the same resolution, 40-minute. They are all derived from the same 10-minute initial time series.*
- *For each of these 4 offset 40min time series, we calculate a set of different metrics. Obviously, we would expect the statistics to be independent of the offset experiment. For illustration, some results are shown for different stations in Figure 2 for standard deviation, autocorrelation at lag-1 and for 5 and 20yrs return levels.*
- *The initial 40min time series (without offset) was next disaggregated to 10min producing 30 time series scenarios.*
- *The same offset experiment is performed for each of the 30 disaggregated time series scenarios. For each scenario, 4 offset 40-minute time series were produced with the 4 different offsets. The process was repeated for each station. The results obtained with the 4 models A, $A^+$, B, $B^+$ are presented in Figure 3 (the MAPE metric is presented for different statistics).*

***The conclusions from our offset experiments are:***
- *Whatever the statistics considered, the estimates calculated on observations for different offsets are very similar. This is highlighted in Figure 2 for 12 stations spread across Switzerland.*
- *When calculated on disaggregated data, the estimates calculated for different offsets are no more similar. More precisely, the estimates obtained for the three non-zero offsets (10, 20, 30-minute) are similar to each other but often significantly different from the reference estimate (with the 0-minute offset). This is highlighted in Figure 3. Each box plot represents the estimated value of a given statistics for 81 stations. Estimates are given for standard deviation (first row), lag-1 correlation (2nd row), and 5 and 10-years return levels (3+4th rows) for the four seasons (the 4 columns) and the different models (model A, B, A+ B+ in the x-axis). Whatever the season, whatever the statistics, the red boxplot (offset 0-min) is very often significantly different from the green/blue/magenta boxplots (10, 20, 30-minute).*
- *Models A and B (without asymmetry) are much more sensitive to temporal offset than models $A^+$ and $B^+$.*
- *The model the less sensitive to temporal offset is model $B^+$.*

[Figure]

*Figure 2.* **Effect of the offset on observed time series statistics.** *Observed metrics as estimated on 40-minute time series obtained for different time axis offsets. On the left is shown the standard deviation and on the right autocorrelation at lag-1. Each panel corresponds to a given station (results presented for 12 stations).*

[Figure]

*Figure 3.* **Effect of the offset on disaggregated time series statistics.** *Mean Absolute Percentage Error (MAPE) between the observed and disaggregated values for different statistics (first row: standard deviation, $2^{nd}$ row: lag-1 autocorrelation, 3 and $4^{th}$ row: 5 and 10-years return level). MAPE is given as a function of the time offset (boxplots of different colors: red - no offset, green, blue, magenta: 10, 20, 30min time offset, respectively), season (DJF, MAM, JJA, SON columns) and disaggregation model (A, $A^+$, B, $B^+$). Each boxplot summarizes the single-site performances obtained for 81 stations spread over Switzerland and for the 40-minute temporal aggregation level.*

*Conclusions: The results of these offset experiments, with observations first and with disaggregated series next, strongly suggest that cascade models that disregard the asymmetry of the cascade generator and its dependency to the asymmetry of the local precipitation definitely break some important precipitation variability features. A comment on this interesting point will be also added in the discussion.*

*Model parsimony argument. One last argument of Rupp et al. (2009) to disregard the asymmetry dependency was the large amount of model parameters required to describe this. This was indeed a critical point of the empirical models of Olsson (1998) and Gunter et al. (2001). Their models are based on empirical ECDFs, which are different from one "asymmetry" class to the other. Their models could thus not easily account for the dependency on intensity (at least not with the scaling relationships of analytical MRC developed by Rupp et al. (2009)). This was indeed an important limitation. This model parsimony argument was also an issue in the analytical model of McIntyre et al. 2016. As mentioned above, they did not consider scaling laws, which are needed to reduce the number of parameters.*

*Our approach fills this gap. We account for the local asymmetry of precipitation in a very parsimonious way, with an asymmetry index which is continuous. Our model can be then analytical, both for the statistical distribution (which is a non-symmetric beta distribution) and for the scaling relationships linking the parameters of the model to different features of the rainfall amount to disaggregate (intensity, asymmetry, temporal scale).*

*With our continuous asymmetry index, we do not have to define classes, allowing a much more robust estimation of model parameters. We are then able to present maps over Switzerland for the 5 parameters of the model. The very large spatial homogeneity of the parameters clearly shows the robustness of the estimates and strongly suggests the relevance of the model with respect to the different features that are of importance for the cascade generator.*

*Is asymmetry an artifact of sampling? Notwithstanding previous elements, from what can be understand from observations, asymmetry in precipitation-related data is not an artifact of sampling.*

- *The asymmetry index we introduce is defined per see. It just characterizes the asymmetry of any given (observed, simulated) temporal rainfall sequence $\{R_{t-1}, R_t, R_{t+1}\}$. As mentioned in the manuscript, for a given time t, the farthest the value of Z is from 0.5, the more asymmetric the sequence is. A Z value close to 0.5 means that $R_{t-1}$ and $R_{t+1}$ are very close to each other, or that they are very small when compared to the amount to disaggregate. This has no link with the sampling artifacts.*
- *As shown in the manuscript, this rainfall sequence asymmetry translates directly to the asymmetry of the ECDF of the breakdown coefficients. This is clear from Figure 3 of the manuscript where statistical characteristics of the breakdown coefficients W are presented as a function of the*

*asymmetry index Z. Some comments will be given in the discussion on these issues.*

**RC1.3** 33-35: Yes, many types and variations of disaggregation models exist. Although it would be excessive to describe them all here, I suggest referencing one or two review papers.

> **AC1.3** *Thank you for this suggestion. There is however to our knowledge no real review paper on disaggregation for the generation of high-resolution data. The review paper of Srikanthan and McMahon (2001) reviews some disaggregation techniques but it is somehow dated and does not consider the disaggregation to sub-daily resolutions. The paper of Koutsoyiannis (2003) gives an interesting and rather large view of different disaggregation techniques but it is not really a review and some approaches are missing (e.g. Method of Fragments). We will nevertheless mention it in the manuscript.*

**RC1.4** 157: Winter should be defined.

> **AC1.4** *As pointed out by you below in RC1.9, it should actually be Autumn which will also be defined as suggested.*

**RC1.5** 169: The text claims to be referring to Figure 1d but I think it should be Figure 1b.

> **AC1.5** *Thank you for pointing out this error. We will fix it.*

**RC1.6** 172: Model B has 5 parameters, not three. I_0 and I_1 should be included in the count of parameters.

> **AC1.6** *We agree that I_0 and I_1 can also be counted as parameters of the model. However, these parameters are kept fixed for all seasons and stations and do not depend on the specific precipitation data at the stations. Here, we will precise that only the free parameters of the model are counted, i.e. the ones that need to be estimated and vary through seasons and stations.*

**RC1.7** 204: Why use lower-case z for the asymmetry index here and below when it was previously upper-case?

> **AC1.7** *We used lower-case in order to point out that z is a realization of the random variable Z. Anyway, we recognise that this can create confusion to the reader so we will uniformize the notation in the whole manuscript.*

**RC1.8** 275-279 & 385-398: Licznar et al. (2011) explore in some detail the artifacts arising from measurement resolution. They present a method of adding small amounts of random noise to discretized observations in an attempt to extract the underlying distribution of W at low intensities. It is at least worth a mention even if the authors don't want to take that approach.

> **AC1.8** *The artifacts arising from measurement resolution lead to critical estimation issues, indeed. Thank you for pointing out this issue here. Actually, during our preliminary analysis, we employed a similar approach of "jittering" observation records. The objective was 1) to understand the influence of rain gauge tipping resolution on the distribution of cascade weights (especially on the non-zero probability amount $p_x$ and on the shape of the distribution for the $W^+$ breakdown coefficient) 2) to determine which data have to be disregarded to allow a relevant estimation of the model parameters, which would be not too much contaminated by the measurement resolution artifact.*

***The jittering process considered in the present study*** *consisted as follows. In Switzerland, sub-daily resolution precipitation is measured with tipping bucket rain gauges, with a measurement resolution of 0.1mm. In the measurement process, one rainfall pulse is recorded once the 0.1mm rainfall bucket is filled. The duration needed to fill the bucket is obviously not known. It may be a few seconds in the case of very intense rainfall rates or several hours or days in the case of very light rainfall rates. Our jittering process is based on the previous consideration. When X mm have been recorded within a given time step (and then displayed in the time series of precipitation data), we considered that:*

- *(X-0.1) mm really belong to this time step and*
- *Only part of the remaining 0.1mm belongs to this time step (this 0.1mm is obviously the first tipping bucket pulse recorded during this time step). The fraction Y of the 0.1 mm belonging to the current time step was sorted randomly. The complementary fraction (1-Y) was attributed to the previous time step.*

***Generation of jittered time series.*** *Our jittering process thus only applies to the first 0.1mm recorded amount (i.e. first recorded tipping bucket pulse) of each time step. It was applied to the original high-resolution time series, available at 10min resolution. We generated 20 jittered time series from the original one and re-estimated the characteristics of the statistical distributions required to describe the breakdown coefficients.*

***Comments from our jittering analyses.***
*As mentioned in many previous papers, in the initial time series, rainfall amounts can only take discrete values. With the 0.1mm resolution, only 0.1, 0.2, 0.3…. 1, 1.1, … values are possible. The probability occurrence of breakdown coefficients with a value of 0 or 0.5, 1/3 or 2/3, 1/4 and 3/4 is overestimated (these values correspond to rainfall amounts of 0.1, or 0.2, or 0.3m, 0.4mm respectively, which are rather frequent). With this jittering process, rainfall values can take all possible positive values and the overestimation mentioned above is largely reduced. We thus largely reduce the artifact due to the tipping bucket resolution. As mentioned in the paper of Licznar et al. (2011b), the statistical distributions of W for the jittered time series are "very smooth and elegant, especially for small time scales ranging from 10 to 80min".*

*These jittering analyses highlighted also the following:*

***Impact of the jittering on precipitation amounts and statistics.*** *By construction, the largest is the initial rainfall amount for a given time step, the lowest is the difference between this initial value and the jittered value. For instance, for a 1mm initial amount, the jittered values can take values between 0.9mm and 1mm. The largest possible difference is 10%. For a 10mm initial amount, the jittered values can take values between 9.9mm and 10mm. The largest possible difference is 1%. The jittering process has thus almost no influence on rainfall properties for moderate to large precipitation amounts. This is for instance the case for precipitation maxima (especially return levels for different return periods) which are almost unchanged.*

***Impact of the jittering on the statistical distribution of the breakdown coefficient.***
*The jittering process leads to significantly modify the distribution of W and, in turn, it can significantly modify the scaling properties of the different parameters of the cascade generator.*

***This was especially highlighted for the non-zero subdivision probability.*** *In the Figure 1a of the manuscript (also shown below in Figure 4), the non-zero subdivision probability $p_x$ varies from 0 - for very low rainfall intensities to 1 - for very large ones.*

- *With the initial series, the precipitation intensity-$p_x$ relationship is found to depend on the temporal scale (see the different curves in Fig. 1a). However, dots reported in the figure for small intensities correspond to small rainfall amounts, which are highly contaminated with the resolution issue as many of these small rainfall amounts are 0.1, 0.2, 0.3 values. This is especially critical for the smallest temporal scales.*

- *When jittered, the differences between the different time scales – mentioned above for instance for the intensity-$p_x$ relationship, is reduced suggesting that an important part of the dependency on temporal scale is mostly due to the measurement precision artifact (see Figure 4a and c). When we disregard all data smaller than 0.8 mm (that are numerous, especially for fine temporal scales), the different dots on the x-axis in the figure with a $p_x$ value of 0 disappear. This confirms the relevance of one unique scaling model for $p_x$ as a function of intensity only. This is the reason why we considered Model B, which by construction disregards possible relationships with temporal scale.*

*This artifact issue obviously deserves more attention in all works with MRC. It would require too long explanation and was not integrated in the manuscript. We will nevertheless mention it more clearly in the discussion.*

(a) (b) (c)

[Figure]

*Figure 4. Non-zero subdivisions probability px as estimated on observation data from Zurich station for each intensity class and temporal scale. (a) Same model A as in the manuscript, (b) model B, a threshold of 0.8mm is applied to discard px (same figure as in Supplementary Material) and (c) same analysis as in panel (a) but this time a jittering procedure is applied to the data before performing the estimation.*

**RC1.9** Table 1: As I stated above Model B has a total of 5 parameters when parameters $I_0$ and $I_1$ are included. Similarly, Model B+ has 7 parameters.

     **AC1.9** *Please see our answer to RC1.6.*

**RC1.10** Figure 1: "SON" should be defined. I assume it is Sep – Nov? Also, the caption says "winter", whereas SON would be autumn.

     **AC1.10** *Thank you for spotting this error. Yes, SON means September to November corresponding to the autumn season. Similarly, DJF means December to February, corresponding to the winter season. This will be clarified.*

**RC1.11** Figure 3: Panel (a) takes effort to interpret. I have a few comments:

    1. What is "x"? Is it W? $W_1$? W+? For clarity, please replace x by what is represent.

    2. An ECDF should go from 0 to 1 but it is not obvious that each individual curve does that. For example, the Z =1 curve appears to have a value F(x) ~= 0.5 at x = 1, but does the Z = 1 curve jump to a value of F(x) = 1 very close to x = 1?

    3. Lastly, although plotting the ECDF is convenient in that several curves can be plotted in one panel, I think it would be much easier to interpret histograms of W for various classes of Z. Notable differences between winter and summer might also be more obvious.

     **AC1.11** *Thank you for the detailed comments on these panels.*

- *x here referred to W. We will replace x by W for simplicity as suggested.*
- *Sure, each ECDF should go from 0 to 1 and it is also the case here. We agree that in the figure this is unnoticeable due to the superposition of different curves and axis. It is also true that for "z=1" the ECDF jumps from around 0.5 for values close to 1, to 1, when x gets 1. The same behaviour can be noticed for "z=0" for very small x. This effect is due to the intermittency of the precipitation process that is reflected on the observed cascade weights with a considerable number of x = 0 or x = 1.*
- *The distribution of positive weights can indeed significantly differ between Winter and Summer and these differences may be slightly difficult to appreciate with ECDFs. Please find below in Figure 4 histograms of observed weights that correspond to different classes of Z-index. On the left are shown histograms for Winter (December to February) and on the right for Summer (June to August). The above assumption is supported by the histograms, even though we believe that the differences can be better observed by looking at the ECDFs than at histograms in our case. It is due to the considerable number of weights equal to 0 or 1, that get more visual attention than the distribution of positive weights, 0<W<1.*

[Figure]

*Figure 4. Histograms of observed breakdown coefficients for each class of asymmetry index Z. In the left panel are the results for winter (DJF) while on the right are the results for summer (JJA). Please note that the weights calculated for precipitation amounts smaller than 0.8 mm are discarded from this analysis.*

**RC1.12** Figure 6: Consider using a log-log scale.

> **AC1.12** *This is a great suggestion as with the current presentation more visual attention is given to very high return levels, which usually occur in summer. Please find below the analogue figure (Figure 5) but in log-log scale. These results correspond to 5-year and 20-year return levels at 40-minute temporal scale. The following representation will replace the one in Figure 6.*

[Figure]

*Figure 5. (**Log-scale**) Observed versus simulated return levels at the 40-minute temporal resolution for (a) 5-year and (b) 20-year return periods, for each model and at each site. Same results as in Figure 6 of the manuscript but log-log scale is used for plotting.*

**RC1.13** 229: Replace "confronted" with "compared".

    **AC1.13** *Thank you for this suggestion. We will account for it.*

**RC1.14** 474: "…reveals actually not obvious…" Typo?

    **AC1.14** *Thank you for pointing out this mistake. We will remove the word "actually".*

**References**

Güntner, A., Olsson, J., Calver, A., and Gannon, B.: Cascade-based disaggregation of continuous rainfall time series: the influence of climate, Hydrol. Earth Syst. Sci., 5, 145–164, https://doi.org/10.5194/hess-5-145-2001, 2001.

Hingray, B. and Ben Haha, M.: Statistical performances of various deterministic and stochastic models for rainfall series disaggregation, Atmos. Res., 77, 152–175, https://doi.org/10.1016/j.atmosres.2004.10.023, 2005.

Koutsoyiannis, D.: Rainfall disaggregation methods: Theory and applications, p. 1–23, https://doi.org/10.13140/RG.2.1.2840.8564, 2003.

Licznar, P., Łomotowski, J., and Rupp, D. E.: Random cascade driven rainfall disaggregation for urban hydrology: An evaluation of six models and a new generator, Atmospheric Research, 99, 563–578, https://doi.org/10.1016/j.atmosres.2010.12.014, 2011b.

Licznar, P., Schmitt, T. G., and Rupp, D. E.: Distributions of microcanonical cascade weights of rainfall at small timescales, Acta Geophys.,59, 1013–1043, https://doi.org/10.2478/s11600-011-0014-4, 2011a.

Olsson, J.: Evaluation of a scaling cascade model for temporal rain- fall disaggregation, Hydrol. Earth Syst. Sci., 2, 19–30, https://doi.org/10.5194/hess-2-19-1998, 1998

Rupp, D. E., Keim, R. F., Ossiander, M., Brugnach, M., and Selker, J. S.: Time scale and intensity dependency in multiplicative cascades for temporal rainfall disaggregation, Water Resour. Res., 45, https://doi.org/10.1029/2008WR007321, 2009.

Srikanthan, R. and McMahon, T. A.: Stochastic generation of annual, monthly and daily climate data: A review, Hydrol. Earth Syst. Sci., 5, 653–670, https://doi.org/10.5194/hess-5-653-2001, 2001.

---

## Author Comment (AC2)

**Response to anonymous referee #2**

*We thank referee #2 for reading the manuscript carefully and providing thoughtful and constructive comments. The comments are noted with **RC**, our responses with **AC**, and the intended additions or changes in the manuscript are underlined.*

**RC2.1** The authors compare their results with a very narrow field of the latest developments (here: analytic solvable micro-canonical cascade models). However, the results can be interpreted from a larger perspective (at least micro-canonical cascade models, better: cascade models in general). For example the scale-dependency (model A vs. B, bounded vs. unbounded model) and position-dependency. The discussion would benefit from it and the reader would be provided with a broader perspective on the scientific field. It is also important because some findings (which are new for the analytic solvable MRC) are quite common to apply for other cascade models.

> **AC2.1** *Thank you for this remark. We agree that our results could be interpreted from a larger perspective for other disaggregation approaches, for other cascade model approaches. We believe that the impact of the local asymmetry of precipitation on precipitation variability will be worth further investigations and may improve our understanding and modelling of some precipitation properties. We will add a comment on this in the conclusion.*

**RC2.2** Section 4.1 The authors highlight the improvement by introducing the asymmetry in comparison to model A and model B. From my understanding neither model A nor model B takes into account the position-dependency of the current time step to disaggregate. To my knowledge the latest references on micro-canonical cascade models all take into account position dependency (so starting, enclosed, ending, isolated position classes depending on the wetness state of: {Rt-1, Rt, Rt+1}). As mentioned before, for analytic solvable MRC it is maybe not common to take into account the positions/patterns from the coarser scale, but it is common for MRC in general. Here, the asymmetry can be interpreted as an extension of the position-dependency, since it takes into account the intensity of the surrounding time steps rather than the wetness state only (so real vs. boolean). So it is not surprising that A+ and B+ outperform A and B, respectively, but A and B do not represent the state-of-science. I recommend to add a position-dependent cascade model to evaluate the added value of the asymmetry in comparison to the wetness state. Even if both approaches result in similar statistics, the asymmetry would have the benefit of being more parameter parsimonious. It is important here to show the reader the clear benefit of the introduced model approach.

> **AC2.2** *Yes, you are right, the asymmetry can be interpreted as an extension of the position-dependency already accounted for in a number of previous works. Models A and B, do not account for the position dependency. They are not the state-of-the-art models if empirical cascade models are considered. They are however state to the art models if analytical models (with scaling models included) are considered. To our knowledge, the gap between both approaches was unsolved to date. Our modelling approach proposes a bridge between both approaches: accounting for asymmetry in an analytical way with a continuous asymmetry index which allows to develop analytical scaling laws.*

*The point you raised is however very relevant. Is there some added value of the asymmetry approach in comparison to the position dependency approach? To assess this, we considered two more models "A position" and "B position" ("Ap" and "Bp"). These two models are based on models A and B respectively, but we added a dependency to the position class (starting, enclosed, ending, isolated). Therefore, for each station, we estimated a set of parameters by season and by position class. In the same manner, as explained in the manuscript, the observed quasi-daily amounts were disaggregated to time series of 40-minute resolution following these two models.*

*Results are shown in Figure 6 and can be interpreted as follows:*
- *For Standard deviation, (a), and the proportion of wet steps, (b): Models Ap / Bp perform similarly as models A/B and models $A^+/B^+$.*
- *For Lag-1, (c), and Lag-2, (d), autocorrelation: Models Ap and Bp perform worse than models A and B (and then much worse $A^+$ and $B^+$)*
- *For mean length of wet spells, (e): Ap is slightly better than A but Ap is still worse than model $A^+$.*
- *For return levels Figure 7, results are almost the same for all models.*

*Overall, in our case, conditioning the analytical models to the position class improves the performance of classical analytical models. As shown in Figure 6 it is however less efficient than considering scaling laws with a continuous asymmetry index. This will be worth further investigations to assess if these results could be valid in other climates. We believe that this better performance of the models $A^+/B^+$ is due to the fact that the models are additionally able to make a distinction between different "starting" sequences (or different ending sequences) as the asymmetry index is also a measure of the "intensity" of the asymmetry (i.e. steep decreasing intensity over the three consecutive precipitation amounts or only slow decreasing) (see $Figure\ 1$ of our response to comment RC1.1). As they are additionally highly parsimonious, Models $A^+$ and $B^+$ appear to be really promising alternative to the class-conditioned models.*

[Figure]

*Figure 6.* **Conditioning the MRC on ending/starting classes.** *Observed versus simulated statistics for each considered model at a 40-minute temporal resolution for different metrics. Four first columns correspond to the results and models presented on the manuscript, the fifth and sixth column correspond to the results obtained when in the model A, respectively model B, the position class dependency is added.*

[Figure]

*Figure 7. **The interest of conditioning the MRC on position classes (ending/starting/enclosed/isolated).** Observed versus simulated return levels at the 40-minute temporal resolution for (a) 5-year and (b) 20-year return periods, for each model and at each site. The four first columns correspond to the results and models presented in the manuscript, the fifth and sixth columns correspond to the results obtained when in model A, respectively model B, the position class dependency is added.*

**RC2.3** The seasonal classification is not common for all cascade models and more common for the pulse models (NSRP & BLRP). The authors show the seasonal variations of parameters in Fig. 8, but I'm still curious how this would affect the results. How would the results look if there is one parameter set applied for the disaggregation of the whole time series?

> **AC2.3** *This is an interesting issue to discuss, thank you for bringing it up. Actually, it not really uncommon for the cascade models to consider a seasonal dependency, see for example Olsson (1998), Günter et al. (2001), Onof et al. (2005), McIntyre et al. (2015). The seasonality of the scaling properties of rainfall – in view of modelling have been also highlighted in other works, especially in Molnar and Burlando (2008) for Switzerland. Conditioning on the season was thus rather natural for us.*
>
> *To show the added value of this seasonal conditioning, we performed the parameter estimation procedure without accounting for seasonality, so only a set of parameters was obtained for each station. In the same manner, as explained in the manuscript, the observed quasi-daily amounts were disaggregated to time series of 40-minute resolution. In order to do a fair comparison with results obtained when considering a seasonal dependency, the evaluation is done on a seasonal basis. The results are not at all satisfactory, neither for standard metrics, see Figure 8, nor for return levels, Figure 9. A huge gap between the results obtained for summer and for other seasons can be noticed, resulting from the mix between short convective events in summer and other precipitation events in other seasons while estimating the parameters.*
>
> *This point will be briefly mentioned in the manuscript.*

[Figure]

*Figure 8. **Results without seasonal stratification** (results to be compared with results of Figure 5 in the manuscript). Observed versus simulated statistics for each considered model at a 40-minute temporal resolution for different metrics. Each triangle represents a site and a season. Same analytical models as in the manuscript but no seasonal stratification is done on parameter estimation.*

[Figure]

*Figure 9.* **Results without seasonal stratification** *(results to be compared with results of Figure 6 in the manuscript). Observed versus simulated return levels at the 40-minute temporal resolution for (a) 5-year and (b) 20-year return periods, for each model and at each site. Same analytical models as in the manuscript but no seasonal stratification is done for parameter estimation.*

**RC2.4** *L2* The term 'simple scaling law' can be confusing. What does 'simple' refer to? Linear? Please clarify.

> **AC2.4** *Thank you for pointing this possible confusing term. We will modify this sentence as follows: This class of models applies scaling models to represent the dependence of the cascade generator on the temporal scale and the precipitation intensity.*

**RC2.5** *L5 '…is usually disregarded.'* I would add the following extension to this sentence: *'…or taken into account in a simplified way.'* (or similar), since there are possibilities out there taking into account the wetness state of the surrounding time steps.

> **AC2.5** *We agree that many empirical cascade models account for the external pattern of precipitation, nevertheless, as far as we are aware this is not the case for analytical scaling models. Our statement was intended for analytical models. For clarification, we will modify the sentence as follows: Although determinant, the dependence on the external precipitation pattern is usually disregarded in the analytical scaling models.*

**RC2.6** L176 The term 'shadow breakdown coefficient' sounds spectacular, but it is not clear what 'shadow' exactly refers to? From my understanding it takes into account the position-dependency as well as the rainfall amounts, because higher rainfall amounts would cause more shadow. However, this name should be introduced/defined so that other authors know when to use it.

> **AC2.6** *Thank you for your comment. The term "shadow breakdown coefficient" will be replaced by "hidden breakdown coefficient".*

**RC2.7** Section 2.2 When introducing Zt the authors could state the intended application briefly and refer to Sec. 2.4 with the detailed description: It only affects p01 and p10, px remains unaffected.

    **AC2.7** *Thank you for the suggestion. We will do so.*

**RC2.8** Fig. 3c For very low and very high values of z tipping points can be identified. How can it be explained? By the measuring resolution of the measuring instrument, leading to minimum values of e.g. 0.1mm?

    **AC2.8** *As mentioned in Section 2.2 of the manuscript, $Z_t$ values close to 0 indicate sequences with very little rain on the first two time steps when compared to the last one (very steep "ascending" sequences), whereas $Z_t$ values close to 1 indicate sequences with very little rain on the two last time steps when compared to the first one (very steep "descending" sequences). This means that the value of the precipitation amount $R_t$ for the central step has to be rather low for those configurations. As a consequence; the observed weights for those rainfall configurations are typically 0 or 1 (this can be observed in the histograms given for the response AC1.11 for the two configurations Z=0.05 and Z=0.95). This leaves only a small sample to estimate the mean of the weights 0<W<1. On the other hand, in order to reduce the effects of measurement artifacts, weights considered for the analysis are only calculated from precipitation amounts $R_t$ larger than 0.8 mm. This is expected to drastically reduce the sample size which we believe is the main reason for the tipping points. This issue will be worth further investigation but as this is not a central issue in our work, we will not discuss/mention it in the manuscript.*

**RC2.9** Table 1-caption. Please add the information that the number of parameters is not taking into account any seasonal variation. So four seasons would lead to 4*parameter number mentioned in the table.

    **AC2.9** *Thank you for your valid suggestion. Such information will be added to the table's caption.*

**RC2.10** L271-274 The description is valid and does not be changed. Nevertheless I'm curious why the authors stop the disaggregation procedure at 40mins and don't go all the way to 10min? Did the scaling behaviour change for finer resolution (often scale invariance hold for ~1d-> ~1h)?

    **AC2.10** *Of course, the disaggregation can go on to finer resolutions. The choice of stopping at 40-minutes was mainly based on the application needs. By contract with the Swiss Confederation, we have to produce weather scenarios for small catchments (from 10 to 1000 $km^2$). The time resolution retained for hydrological simulation in the project was thus 30min. On the other hand, the artifact induced by the measurement precision would make difficult the evaluation of disaggregated scenarios at lower temporal resolutions (e.g. Paschalis et al. (2013)).*

**RC2.11** Section 2.6 Maybe I've just not seen it: Which distribution function is used to estimate the return periods analysed in Fig. 6?

    **AC2.11** *Actually, no distribution is fitted to the data. We use the Gringorten plotting position to plot annual maximums, and then the quantile for a given non-exceedance probability is determined by linear interpolation of annual maxima. This will be precised in the revised manuscript.*

**RC2.12** L379-384 The scale-dependency often plays a minor role if the scaling behaviour is linear, which is often the case for the analysed range of resolution in this study. I suggest

to add a figure on scaling behaviour (the typical Mq (Moments of order q=1,2,3,..)-temporal resolution-plot) to verify the finding that A not necessarily outperforms B. Other cascade models apply scale-invariance already for this range of temporal resolutions (e.g. Günther et al, 2001)

**AC2.12** *Please find below in Figure 10 the moments of order 1 to 4 estimated on observation data (points) and on the generated scenarios (the lines show the median among the metrics estimated on 30 scenarios). The results here concern the station of Zurich. Each column corresponds to a model and each row to a season. We find the differences between models are only minor for moments of order 1 to 3, while for 4-order moments more differences can be noticed and depend on the season. To our opinion, this interesting scaling behaviour of precipitation is rather out of scope of our study. We will therefore not mention it in the revised version.*

[Figure]

Figure 10. Log-log plots of the empirical q moments versus temporal scale. Each color corresponds to one moment, dots to moments estimated on observations, and full lines correspond the median of estimated moments on the 30 scenarios. Each column to one model and each row to one season. The analysis is performed on the data for Zurich station.

**RC2.13** Fig. 5a) 'Standard deviation' – of what?

**AC2.13** *Thank you for noting that the description needs to be detailed. We will add "Standard deviation of precipitation".*

**RC2.14** *L376-378 This is maybe true for analytical developments, but for non-analytical approaches position-dependencies are most often taken into account. This information should be added here for the sake of completeness.*

**AC2.14** *Our focus here was on the analytical scaling models. Anyway, this is a valid point and we will complete this information as suggested.*

**RC2.15** *L408-420 The persistence/intermittency is a weakness of micro-canonical cascade models. Müller-Thomy (2020) has introduced an extended position-dependency that improves the autocorrelation for all lags. Here, lag-1 and lag-2 are studied, results for other lags are not shown. Are results similar for all lags? Would the involvement of the extended position-dependency (would be an extended asymmetry approach) also an (additional) improvement for the analytical MRC?*

**AC2.15** *Yes, results are similar for all lags. We had mentioned it in the manuscript. Figure 11 below presents results obtained for other lags (lag-3 to lag-6 estimated on 40-minute data). Accounting for the asymmetry index significantly improves the reproduction of the lag 1 and 2. The added value decreases with higher degree lags but is still important for lag 3 and 4 (see Figure 11 below).*

*Thank you also for the interesting question relative to the "Extended position dependency". We have shown in our paper that the asymmetry of the CDF depends on the precipitation structure of the precipitation sequence $\{R_{t-1}, R_t, R_{t+1}\}$ and that the hidden breakdown coefficient Z is a skilful predictor for this dependence. We could also expect that the CDF asymmetry additionally depends on the precipitation structure at different coarser temporal resolutions and that $n^{th}$ order hidden breakdown coefficient $Z_n$ defined with the extended precipitation sequence $\{R_{t-n}, \ldots, R_{t-1}, R_t, R_{t+1}, \ldots R_{t+n}\}$ could be of interest there. This will however introduce some additional complexity to the model. This will be worth further investigation.*

**RC2.16** Fig. 9 Which temporal resolution is shown here?

**AC2.16** *Thank you for pointing out that the specification of the temporal resolution is missing for the Figure 9a. We will specify that the temporal resolution is 40-minutes.*

[Figure]

*Figure 11. Observed versus simulated statistics for each considered model at a 40-minute temporal resolution for lags of higher order.*

**References**

Gringorten, I. I.: A plotting rule for extreme probability paper, J. Geophys. Res., 68, 813–814, https://doi.org/10.1029/JZ068i003p00813, 1963

Güntner, A., Olsson, J., Calver, A., and Gannon, B.: Cascade-based disaggregation of continuous rainfall time series: the influence of climate, Hydrol. Earth Syst. Sci., 5, 145–164, https://doi.org/10.5194/hess-5-145-2001, 2001.

McIntyre, N., Shi, M., and Onof, C.: Incorporating parameter dependencies into temporal downscaling of extreme rainfall using a random cascade approach, J. Hydrol., 542, 896–912, https://doi.org/10.1016/j.jhydrol.2016.09.057, 2016.

Molnar, P. and Burlando, P.: Variability in the scale properties of high-resolution precipitation data in the Alpine climate of Switzerland:
VARIABILITY IN SCALE PROPERTIES, Water Resour. Res., 44,
https://doi.org/10.1029/2007WR006142, 2008.

Onof, C., Townend, J., Kee, R., 2005. Comparison of two hourly to 5-min rainfall disaggregators. Atmos. Res. 77 (1–4), 176–187.

Paschalis, A., P. Molnar, and P. Burlando (2012), Temporal dependence structure in weights in a multiplicative cascade model for precipitation, Water Resour. Res., 48, W01501, doi:10.1029/2011WR010679.

---

## Referee Report (RR1)

This was the second time I was involved as a reviewer for this manuscript. As I have mentioned before, the manuscript is well-written and I enjoyed reading it. My comments of the previous round of reviews have almost all been solved, only two minor points still have to be handled. I also want to emphasize that the authors satisfied me with their review in a very positive way. Additional model runs and new model versions were analysed for the reply, although some of them were beyond the scope of the study (as the authors state). This is a very constructive reply and the scientific community benefits from such efforts!

- Hannes Müller-Thomy

Comments from previous review:

RC2.7 Section 2.2 When introducing Zt the authors could state the intended application briefly and refer to Sec. 2.4 with the detailed description: It only affects $p01$ and $p10$, $px$ remains unaffected.
AC2.7 Thank you for the suggestion. We will do so.

-> I could not find any modification in the manuscript to this point.

RC2.13 Fig. 5a) 'Standard deviation' – of what?

AC2.13 Thank you for noting that the description needs to be detailed. We will add "Standard deviation of precipitation".

-> 'Standard deviation of precipitation' is not concise and can represent various characteristics, I suggest 'Standard deviation of precipitation intensity'.

---

## Author Response (AR2)

Dear editor and referees,

Thank you very much for the reassessment of our manuscript and for the interesting complementary comments. Concerning the relevant suggestion made by referee #1 to show the effect of offset on the breakdown coefficients of disaggregated data, we have done some supplementary analysis, which revealed interesting results. Therefore, we have added two figures in the Supplementary Material and two relative comments in section 5.2 about these results.

Best regards,

Kaltrina Maloku

**Response to anonymous referee #1**

*We thank referee #1 for carefully reassessing the revised manuscript and for the constructive comments. The comments are noted with RC and our responses with AC.*

**RC1.1** The authors demonstrate that resampling the disaggregated time series with a constant offset clearly affects time series statistics (Figure S11). While overall the effect of offsetting appears smaller for asymmetric models, the effects are not negligible for asymmetric models, particularly when considering autocorrelation. Curiously, offsetting appears to increase error in some cases and decrease it in others. Have the authors considered why this is?

> **AC1.1** *In Figure S11 we first wanted to point out the stability of the statistics when they are estimated with different offsets. All in all, stability is better when the asymmetry dependency is accounted for in the cascade generator. We acknowledge that this is not clear for model A+ when considering the autocorrelation. At this point, we cannot give a precise explanation to this result. It may be partly due to the boxplot representation (the number of points shown as outliers or included otherwise in the boxplot) or to the fact that the disaggregation to 10min uses a cascade generator estimated on larger temporal scales (for the 10 and 20min scales, we use the scaling laws obtained for scales > 40min). This will need further investigation.*
> *However, note that better stability is obtained whatever the statistics for model B+. Except for autocorrelation, a significant improvement in the stability is also obtained with model A+ when compared to models A and B. This is especially the case for the ECDFs of the breakdown coefficients. See our response to comment RC1.2 below.*

**RC1.2** Instead of comparing statistical error, the matter could be addressed by more directly examining the distribution of the BDCs recreated from the disaggregated data, before and after offsetting (as in Rupp et al. 2009). I request that the authors show effect of offsetting on the BDCs of the disaggregated data, either through histograms such as those on page 13 of the authors' response (egusphere-2023-544-author_response-version1.pdf), or through the effect on the distribution of the asymmetry index Z.

**AC1.2** *Thank you for this very relevant suggestion. We have done this analysis and it is indeed very conclusive. Therefore, we have added two figures in the Supplementary Material and two relative comments in section 5.2 about the results. These results have also been recalled in the conclusion.*
*The two figures added to the SM (New Figures S12 and S13) are given below. Basically, the suggested analysis even better highlights the interest in including precipitation asymmetry in the cascade generator. It allows us to show the two following additional results:*

- *As shown in Figure 3 of the manuscript, the statistical distribution of BDCs is expected to strongly depend on the external pattern of precipitation and more precisely on the value of the asymmetry index Z. When the asymmetry index Z is larger, the probability p01 (resp. p10) is expected to be smaller (resp. larger) and the mean of the distribution is expected to be larger (cf. ECDF of BDCs for Zurich data in Figure 3). As illustrated in Figure S12 and S13 of SM, the dependency of ECDF of BDCs on Z is actually almost perfectly reproduced with models A+ and B+, when precipitation asymmetry is accounted for in the cascade generator. Conversely, it is fully missed in models A and B: with models A and B, the dependence of the ECDF of BDCs on Z seems to fully disappear and the ECDFs turn to be always symmetric. It is not a surprise, owing to the symmetric formulation of the cascade generator in both models.*

- *On the other hand, the statistics of a precipitation time series are not expected to change when the time series is offset in time by a small time duration. This should be the case for standard statistics, and as suggested by Reviewer1 this should be also the case for the ECDFs of the BDCs.*

    1. *This is indeed the case, when the offset experiment is applied on observed data: the statistics, estimated on the four offset time series respectively, are similar. This was illustrated in the previous manuscript version in Figure S10 for standard precipitation statistics of 12 stations. As expected, this is also the case for the ECDFs of the observed BDCs as illustrated for Zurich data in figures S12 and S13.*

    2. *As mentioned in our manuscript, the offset-independence property is conversely not necessarily satisfied when the offset experiment is applied to the times series generated with the disaggregation models. When calculated on disaggregated data, the estimates calculated for different offsets are no longer necessarily similar. In general, the models A and B (without asymmetry) are much more sensitive to temporal offset than models A+ and B+: This was illustrated for standard statistics as highlighted in Figure S11 of the SM. This is even worse for the ECDFs of BDCs as illustrated in Figures S12 and S13 of the SM. Including the asymmetry in the*

*generator allows to almost fully correct this limitation, making the ECDFs rather insensitive to the offset.*

[Figure]

*New Figure S12. Empirical cumulative distribution function of observed and simulated BDCs by each model and with different offsets (first line : no offset, 2nd, 3rd, 4th lines : 10min, 20, min, 30min offsets). The ECDFs are plotted for different classes of the asymmetry index, as in Figure 3 of the manuscript. The first lines show the ability of the model to reproduce (Model A+ and B+) and the inability to reproduce (Model A and B) the dependency of the ECDF to the Z class. The ECDFs obtained with model A and B are all symmetric, contrary to what is observed. Winter season.*

[Figure]

*New Figure S13. Same description but for summer season.*

**RC1.3** While the authors have done an admirable amount of additional analysis, I do not think they have yet shown strong evidence that refutes the "suspicion" given by Rupp et al. (2009), although they claim in their response that this suspicion was likely wrong.

It may well not be (and need not be) that the authors' paper is the one the provides a definitive answer to the question. Still, I think the authors downplay the issue in their Discussion given apparent asymmetry is so central to their approach. It would be more appropriate to acknowledge that this remains an unresolved issue.

> **AC1.3** *We believe that the Rupp et al. suspicion was likely wrong or at least not as critical as they suggest. However, we acknowledge that additional analyses are required to confirm our results. We have adapted our text accordingly.*
>
> > • *In the text (ln 415): "We argue that a large part of the dependency on the temporal scale is also a result of this precision artefact. This is clearly*

*shown here concerning the probability $p_x(I)$." > we changed "clearly shown" by "strongly suggested"*

- *We have also added line 420 "Further investigations will be worth to assess if this would still hold in other contexts worldwide."*

**RC1.4** L77: Replace "worth to be mentioned" with "worth mentioning."

    **AC1.4** *Thank you for the suggestion. It has been accounted for.*

**RC1.5** L122: The authors don't need a reference for the Beta distribution itself. If they to cite McIntyre et al. (2016) as a case where the 2-parameter Beta distribution was used to characterize the distribution of W, is McIntyre the earliest example of this? If not, they could instead cite a first or early use of it in this context (if known) or, at minimum, put "e.g." before McIntyre et al.

    **AC1.5** *As far as we are aware this study is one the earliest examples of the use of two-parameter Beta distribution to model the distribution of W. Nevertheless, we added e.g. before McIntyre et al. as suggested.*

**RC1.6** L189: Replace "is the asymmetry of the sequence" with "the asymmetry of the sequence is".

    **AC1.6** *It has been done.*

**RC1.7** L203-204: Why is "increasing" called "right valley" if the low point is to the left? Similarly, with "left valley". Am I misunderstanding something?

    **AC1.7** *The sentence formulation is a priori right. 6 external patterns are sometimes defined, as follows (fig 1 of Hingray et Ben Haha (2005), following Ormsbee (1989)):*

[Figure]

Fig. 1. Geometric similarity between external and internal temporal rainfall patterns for the six pattern classes according to Ormsbee (1989). The internal temporal pattern determined by the time evolution $g(t)$ of rainfall intensity is proportional to the piece wise linear function $R_i^* \cdot g(t)$ plotted on the schemes. The rainfall volume of each 10-min bar is derived from $g(t)$ via Eq. (7).

*In the classes "descending", "left valley" and "left peak", Rt-1 > Rt+1*
*In the classes "ascending", "right valley" and "right peak", Rt-1 < Rt+1*
*This is coherent with what is written in lines L203-204 :*

*"$Z_t<0.5$ indicates an "increasing" or "right valley" sequence, i.e. a sequence where $R_{t-1} < R_{t+1}$, while $Z_t>0.5$ indicates a "decreasing" or "left valley" sequence ($R_{t-1} > R_{t+1}$).*

**RC1.8** L325: Replace "20 years return period" with "20-year return period".

L325: Replace the "The 5-year, respectively 20-year, return level" with "The 5 and 20-year return level, respectively".

L327: Replace "5-year, respectively 20-year", with "5 and 20 years, respectively."

**AC1.8** *Thank you very much for the welcomed suggestions. This has been modified.*

**RC1.9** L395-396: Replace "thanks to" with "by".

**AC1.9** *This has been modified.*

**RC1.10** L422: I believe the actual Licznar et al. (2011) paper that applied the random perturbations was: "Licznar, P., Schmitt, T. G., & Rupp, D. E. (2011). Distributions of microcanonical cascade weights of rainfall at small timescales. Acta Geophysica, 59, 1013-1043" although this paper is not listed in the References.

**AC1.10** *Thank you for noticing this error. We now cite the correct reference in the revised manuscript.*

**RC1.11** L428-429: I don't understand what "further leading on removal of parts" means. Please rephrase.

**AC1.11** *The dependency of W distribution to the temporal aggregation level for small intensities is partly due to the precision artefact as discussed above in the same section. In Figure 4 of the authors' response to Reviewer 1, we show how the dependency to the temporal scale reduces after employing the jittering process since over-represented values of BDC W = 1/2, 1/3,…, are partly removed. We will rephrase as follows: "This leads in turn part of the "scaling dependencies" mentioned above to disappear".*

**RC1.12** L454 and elsewhere: Sometimes "Supplementary Material" is written and sometimes simply "SM". Please be consistent.

**AC1.12** *Thank you for noting this inconsistency. We now write "Supplementary Material" when referred to it for the first time in Section 4.1, and "SM" otherwise.*

**RC1.13** How were the disaggregated 40-minute time series further disaggregated to 10 minutes? Using the MRC models? Or was each 10-minute interval allotted 25% of 40-minute rainfall total?

**AC1.13** *The 40-minute disaggregated time series were further disaggregated to 10-minute time series using the same MRC models used to disaggregate to 40-minute time series. To get the parameters of the generator for the high resolutions, we use the scaling relationships identified from the lower resolutions. We now precise this information in the revised manuscript.*

**RC1.14** L461: Why "thirty" 10-minute time series?

    *AC1.14 The disaggregation to 40-minute resolution is performed thirty times as mentioned in Section 2.6, thus thirty 40-minute time series are obtained by each MRC model and station. Each of these scenarios is further disaggregated into a 10-minute time series, resulting in thirty 10-minute time series by model and station.*

**RC1.15** L463: I think "no more similar" should be "no longer similar".

    *AC1.15 The paragraph has been rewritten to include new discussions (see RC1.2 above).*

**RC1.16** L468: Replace "accounted for for a long time" with "considered for a long time".

    *AC1.16 Thank you for your suggestion. It is modified.*

**Response to anonymous referee #2**

*We thank referee #2 for carefully reassessing the revised manuscript and for the very encouraging comments. The comments are noted with RC and our responses with AC.*

**RC2.1** RC2.7 Section 2.2 When introducing Zt the authors could state the intended application briefly and refer to Sec. 2.4 with the detailed description: It only affects p01 and p10, px remains unaffected.

AC2.7 Thank you for the suggestion. We will do so.

-> I could not find any modification in the manuscript to this point.

**AC2.1** *Thank you for this comment. Yes, px remains unaffected. It affects p01 and p10, and in addition, it also affects the mean of the distribution of the positive cascade weights (0<W<1). In this section, we refer to these details to the following sections. In line 197 we have added: Details about the use of this index in the cascade generator are given in the following sections.*

**RC2.2** RC2.13 Fig. 5a) 'Standard deviation' – of what?

AC2.13 Thank you for noting that the description needs to be detailed. We will add "Standard deviation of precipitation".

-> 'Standard deviation of precipitation' is not concise and can represent various characteristics, I suggest 'Standard deviation of precipitation intensity'.

**AC2.2** *As the time series represent precipitation amounts, we have added "Standard deviation of precipitation amounts".*